# Generalizing Stochastic Smoothing for Differentiation and Gradient Estimation

## Abstract

We deal with the problem of gradient estimation for stochastic differentiable relaxations of algorithms, operators, simulators, and other non-differentiable functions. Stochastic smoothing conventionally perturbs the input of a non-differentiable function with a differentiable density distribution with full support, smoothing it and enabling gradient estimation. Our theory starts at first principles to derive stochastic smoothing with reduced assumptions, without requiring a differentiable density nor full support, and presenting a general framework for relaxation and gradient estimation of non-differentiable black-box functions $f : \mathbb{R}^n \to \mathbb{R}^m$. We develop variance reduction for gradient estimation from 3 orthogonal perspectives. Empirically, we benchmark 6 distributions and up to 24 variance reduction strategies for differentiable sorting and ranking, differentiable shortest-paths on graphs, differentiable rendering for pose estimation, as well as diff. cryo-ET simulations.

## 1 Introduction

The differentiation of algorithms, operators, and other non-differentiable functions has been a topic of rapidly increasing interest in the machine learning community [1]–[7]. In particular, whenever we want to integrate a non-differentiable operation (such as ranking) into a machine learning pipeline, we need to relax it into a differentiable form in order to allow for backpropagation. To give a concrete example, a body of recent work considered continuously relaxing the sorting and ranking operators for tasks like learning-to-rank [5], [7]–[15]. These works can be categorized into either casting sorting and ranking as a related problem (e.g., optimal transport [16]) and differentiably relaxing it (e.g., via entropy-regularized OT [7]) or by considering a sorting algorithm and continuously relaxing it on the level of individual operations or program statements [5], [11], [14], [15]. To give another example, in the space of differentiable graph algorithms and clustering, popular directions either relax algorithms on a statement level [5] or cast the algorithm as a convex optimization problem and differentiate the solution of the optimization problem under perturbed parameterization [2], [3], [17].

Complementary to these directions of research, in this work, we consider algorithms, operators, simulators, and other non-differentiable functions directly as black-box functions and differentiably relax them via stochastic smoothing [18], i.e., via stochastic perturbations of the inputs and via multiple function evaluation. This is challenging as, so far, gradient estimators came with large variance and supported only a restrictive set of smoothing distributions. More concretely, for a black-box function $f : \mathbb{R}^n \to \mathbb{R}^m$, we consider the problem of estimating the derivative (or gradient) of the relaxation

$$f_\epsilon(x) = \mathbb{E}_{\epsilon \sim \mu}\big[f(x + \epsilon)\big] = \int f(x + \epsilon)\,\mu(\epsilon)\,\mathrm{d}\epsilon \tag{1}$$

where $\epsilon$ is a sample from a probability distribution with an (absolutely) continuous density $\mu(\epsilon)$. $f_\epsilon$ is a differentiable function (regardless of differentiability properties of $f$ itself, see Section 2 for details) and its gradient is well defined. Under limiting restrictions on the probability distribution $\mu$ used for smoothing, gradient estimators exist in the literature [2], [18]–[20].

The **contribution** of this work lies in providing more generalized gradient estimators (reducing assumptions on $\mu$, Lemma 3) that exhibit reduced variances (Sec. 2.2) for the application of differentiably relaxing conventionally non-differentiable algorithms. Moreover, we enable smoothing with and differentiation wrt. non-diagonal covariances (Thm. 7), characterize formal requirements for $f$ (Lem. 8+9), discriminate smoothing of algorithms and losses (Sec. 2.3), and provide a $k$-sample median extension (Apx. C). The proposed approach is applicable for differentiation of (i) arbitrary[1]

functions, which are (ii) considered as a black-box, which (iii) can be called many times (primarily) at low cost, and (iv) should be smoothed with any distribution with absolutely continuous density on $\mathbb{R}$. This contrasts prior work, which smoothed (i) convex optimizers [2], [3], (ii) used first-order gradients [7], [11], [15], [21], (iii) allowed calling an environment only once or few times in RL [20], and/or (iv) smoothed with fully supported differentiable density distributions [2], [3], [18], [22]. In machine learning, many other fields also utilize the ideas underlying stochastic smoothing; stochastic smoothing and similar methods can be found, e.g., in REINFORCE [20], the score function estimator [23], the CatLog-Derivative trick [24], perturbed optimizers [2], [3], among others.

## 2 DIFFERENTIATION VIA STOCHASTIC SMOOTHING

We begin by recapitulating the core of the stochastic smoothing method. The idea behind smoothing is that given a (potentially non-differentiable) function $f : \mathbb{R}^n \to \mathbb{R}^m$ [1], we can relax the function to a differentiable function by perturbing its argument with a probability distribution: if $\epsilon \in \mathbb{R}^n$ follows a distribution with a differentiable density $\mu(\epsilon)$, then $f_\epsilon(x) = \mathbb{E}_\epsilon[f(x + \epsilon)]$ is differentiable.

For the case of $m = 1$, i.e., for a scalar function $f$, we can compute the gradient of $f_\epsilon$ by following and extending part of Lemma 1.5 in Abernethy *et al.* [18] as follows:

**Lemma 1** (Differentiable Density Smoothing). *Given a function $f : \mathbb{R}^n \to \mathbb{R}$ [1] and a differentiable probability density function $\mu(\epsilon)$ with full support on $\mathbb{R}^n$, then $f_\epsilon$ is differentiable and*

$$\nabla_x f_\epsilon(x) = \nabla_x \mathbb{E}_{\epsilon \sim \mu}\big[f(x + \epsilon)\big] = \mathbb{E}_{\epsilon \sim \mu}\big[f(x + \epsilon) \cdot \nabla_\epsilon - \log \mu(\epsilon)\big] . \tag{2}$$

*Proof.* For didactic reasons, we include a full proof in the paper to support the reader's understanding of the core of the method. Via a change of variables, replacing $x + \epsilon$ by $u$, we obtain ($d\epsilon/du = 1$)

$$f_\epsilon(x) = \int f(x + \epsilon)\mu(\epsilon) \, d\epsilon = \int f(u)\mu(u - x) \, du . \tag{3}$$

Now,

$$\nabla_x f_\epsilon(x) = \nabla_x \int f(u)\mu(u - x) \, du = \int f(u) \, \nabla_x \mu(u - x) \, du . \tag{4}$$

Using $\nabla_x \mu(u - x) = -\nabla_\epsilon \mu(\epsilon)$,

$$\nabla_x f_\epsilon(x) = - \int f(x + \epsilon) \, \nabla_\epsilon \mu(\epsilon) \, d\epsilon . \tag{5}$$

Because $\frac{\partial \mu(\epsilon)}{\partial \epsilon} = \mu(\epsilon) \cdot \frac{\partial \log \mu(\epsilon)}{\partial \epsilon}$, we can simplify the expression to

$$\nabla_x f_\epsilon(x) = - \int f(x + \epsilon) \, \mu(\epsilon) \, \nabla_\epsilon \log \mu(\epsilon) \, d\epsilon = \mathbb{E}_{\epsilon \sim \mu}\left[f(x + \epsilon) \, \nabla_\epsilon - \log \mu(\epsilon)\right] . \tag{6}$$

$\square$

Empirically, for a number of samples $s$, this gradient estimator can be evaluated without bias via

$$\nabla_x f_\epsilon(x) \triangleq \frac{1}{s} \sum_{i=1}^{s} \left[f(x + \epsilon_i) \, \nabla_{\epsilon_i} - \log \mu(\epsilon_i)\right] \qquad \epsilon_1, ..., \epsilon_s \sim \mu . \tag{7}$$

**Corollary 2** (Differentiable Density Smoothing for Vector-valued Functions). *We can extend Lemma 1 to vector-valued functions $f : \mathbb{R}^n \to \mathbb{R}^m$, allowing to compute the Jacobian matrix $\mathbf{J}_{f_\epsilon} \in \mathbb{R}^{m \times n}$ as*

$$\mathbf{J}_{f_\epsilon}(x) = \mathbb{E}_{\epsilon \sim \mu}\left[f(x + \epsilon) \cdot \left(\nabla_\epsilon - \log \mu(\epsilon)\right)^\top\right] . \tag{8}$$

We remark that prior work (e.g., [18]) limits $\mu$ to be a differentiable density with full support on $\mathbb{R}$, typically of exponential family, whereas we generalize it to any absolutely continuous density, and include additional generalizations. This has important implications for distributions such as Cauchy, Laplace, and Triangular, which we show to have considerable practical relevance.

---

[1]Traditionally and formally, only functions with compact range have been considered for $f$ (i.e., $f : \mathbb{R}^n \to [a, b]$) [19], [25]. More recently, i.a., Abernethy *et al.* [18] have considered the general case of function with the real range. While this is very helpful, e.g., enabling linear functions for $f$, this is (even without our generalizations) not always finitely defined as we discuss with the help of degenerate examples in Appendix B. There, we characterize the set of valid $f$ leading to finitely defined $f_\epsilon$ in Lemma 8 as well as $\nabla f_\epsilon$ in Lemma 9 in dependence on $\mu$. We remark that, beyond discussions in Appendix B, we assume this to be satisfied by $f$.

**Lemma 3** (Requirement of Continuity of $\mu$). *If $\mu(\epsilon)$ is absolutely continuous (and not necessarily differentiable), then $f_\epsilon$ is continuous and differentiable everywhere.*

$$\nabla_x f_\epsilon(x) = \mathbb{E}_{\epsilon \sim \mu} \left[ f(x + \epsilon) \cdot \mathbf{1}_{\epsilon \notin \Omega} \cdot \nabla_\epsilon - \log \mu(\epsilon) \right]. \tag{9}$$

*$\Omega$ is the zero-measure set of points with undefined gradient. We provide the proof in Appendix A.1.*

Lemma 3 has important implications. In particular, it enables smoothing with non-differentiable density distributions such as the Laplace distribution, the triangular distribution, and the Wigner Semicircle distribution [26], [27] while maintaining differentiability of $f_\epsilon$.

**Remark 4** (Requirement of Continuity of $\mu$). However, it is crucial to mention that, for stochastic smoothing (Lemmas 1, 3, Corollary 2), $\mu$ *has to be continuous*. For example, the uniform distribution is *not* a valid choice because it does not have a continuous density on $\mathbb{R}$. ($\mathcal{U}(a, b)$ has discontinuities at $a, b$ where it jumps between $0$ and $1/(b-a)$.) With other formulations, e.g., [28], [29], it is possible to perform smoothing with a uniform distribution over a ball; however, if $f$ is discontinuous, uniform smoothing may not lead to a differentiable function. Continuity is a requirement but not a sufficient condition, and absolutely continuous is a sufficient condition; however, the difference to continuity corresponds only to non-practical and adversarial examples, e.g., the Cantor or Weierstrass functions.

**Remark 5** (Gaussian Smoothing). A popular special case of differentiable stochastic smoothing is smoothing with a Gaussian distribution $\mu_\mathcal{N} = N(\mathbf{0}_n, \mathbf{I}_n)$. Here, due to the nature of the probability density function of a Gaussian, $\nabla_\epsilon - \log \mu_\mathcal{N}(\epsilon) = \epsilon$. Further, when $\mu_{\mathcal{N}_\sigma} = N(\mathbf{0}_n, \sigma^2 \mathbf{I}_n)$, then $\nabla_\epsilon - \log \mu_{\mathcal{N}_\sigma}(\epsilon) = \epsilon/\sigma$. We emphasize that this equality *only* holds for the Gaussian distribution.

Equipped with the core idea behind stochastic smoothing, we can differentiate any function $f$ via perturbation with a probability distribution with (absolutely) continuous density.

Typically, probability distributions that we consider for smoothing are parameterized via a scale parameter, viz., the standard deviation $\sigma$ in a Gaussian distribution or the scale $\gamma$ in a Cauchy distribution. Extending the formalism above, we may be interested in differentiating with respect to the scale parameter $\gamma$ of our distribution $\mu$. This becomes especially attractive when optimizing the scale and, thereby, degree of relaxation of our probability distribution. While our formalism allows reparameterization to express $\gamma$ within $\mu$, we can also explicitly write it as

$$\nabla_x f_{\gamma\epsilon}(x) = \nabla_x \mathbb{E}_{\epsilon \sim \mu} \left[ f(x + \gamma \cdot \epsilon) \right] = \mathbb{E}_{\epsilon \sim \mu} \left[ f(x + \gamma \cdot \epsilon) \cdot \left( \nabla_\epsilon - \log \mu(\epsilon) \right) / \gamma \right]. \tag{10}$$

Now, we can differentiate wrt. $\gamma$, i.e., we can compute $\nabla_\gamma f_{\gamma\epsilon}(x)$.

**Lemma 6** (Differentiation wrt. $\gamma$). *Extending Lemma 1, Corollary 2, and Lemma 3, we have*

$$\nabla_\gamma f_{\gamma\epsilon}(x) = \nabla_\gamma \mathbb{E}_{\epsilon \sim \mu} \left[ f(x + \gamma \cdot \epsilon) \right] = \mathbb{E}_{\epsilon \sim \mu} \left[ f(x + \gamma \cdot \epsilon) \cdot \left( -1 + \left( \nabla_\epsilon - \log \mu(\epsilon) \right)^\top \cdot \epsilon \right) / \gamma \right]. \tag{11}$$

*The proof is deferred to Appendix A.2.*

We can extend $\gamma$ for multivariate distributions to a scale matrix $\mathbf{\Sigma}/\mathbf{L}$ (e.g., a covariance matrix). This enables optimization for finding the optimal scale matrix $\mathbf{L}$, and further not only isotropic distributions, but instead also, e.g., multivariate Gaussians with correlations between dimensions.

**Theorem 7** (Multivariate Smoothing with Covariance Matrix). *We have a function $f : \mathbb{R}^n \to \mathbb{R}^m$. We assume $\epsilon$ is drawn from a multivariate distribution with absolutely continuous density in $\mathbb{R}^n$. We have an invertible scale matrix $\mathbf{L} \in \mathbb{R}^{n \times n}$ (e.g., for a covariance matrix $\mathbf{\Sigma}$, $\mathbf{L}$ is based on its Cholesky decomposition $\mathbf{L}\mathbf{L}^\top = \mathbf{\Sigma}$). We define $f_{\mathbf{L}\epsilon}(x) = \mathbb{E}_{\epsilon \sim \mu} \left[ f(x + \mathbf{L} \cdot \epsilon) \right]$. Then, our derivatives $\partial f_{\mathbf{L}\epsilon}(x) / \partial x$ $(\in \mathbb{R}^{m \times n})$ and $\partial f_{\mathbf{L}\epsilon}(x) / \partial \mathbf{L}$ $(\in \mathbb{R}^{m \times n \times n})$ can be computed as*

$$\nabla_x \left( f_{\mathbf{L}\epsilon}(x) \right)_i = \mathbb{E}_{\epsilon \sim \mu} \left[ f(x + \mathbf{L} \cdot \epsilon)_i \cdot \mathbf{L}^{-1} \cdot \left( \nabla_\epsilon - \log \mu(\epsilon) \right) \right], \tag{12}$$

$$\nabla_\mathbf{L} \left( f_{\mathbf{L}\epsilon}(x) \right)_i = \mathbb{E}_{\epsilon \sim \mu} \left[ f(x + \mathbf{L} \cdot \epsilon)_i \cdot \mathbf{L}^{-\top} \cdot \left( -1 + \left( \nabla_\epsilon - \log \mu(\epsilon) \right) \cdot \epsilon^\top \right) \right]. \tag{13}$$

*Above, the indicator (from* (9)*) is omitted for a simplified exposition. Proofs are deferred to Apx. A.3.*

In Appendix C, we additionally extend stochastic smoothing to differentiating the expected $k$-sample median, show that it is differentiable, and provide an unbiased gradient estimator in Lemma 12.

Table 1: Probability distributions considered for generalized stochastic smoothing. Displayed is (from left to right) the density of the distribution $\mu(\epsilon)$ (plot + equation), the derivative of the NLL (equation), and the product between the density and the derivative of the NLL (plot). The latter plot corresponds to the kernel that $f$ is effectively convolved by to estimate the gradient. (∗): applies to $\epsilon \in (-1, 1) \setminus \{0\}$, otherwise 0 or undefined.

| Distribution | Density / PDF $\mu(\epsilon)$ | $\nabla_\epsilon - \log \mu(\epsilon)$ | $\mu(\epsilon) \cdot \nabla_\epsilon - \log \mu(\epsilon)$ |
|---|---|---|---|
| Gaussian | $\dfrac{1}{\sqrt{2\pi}} \exp\left( -\nicefrac{1}{2} \cdot \epsilon^2 \right)$ | $\epsilon$ | |
| Logistic | $\dfrac{\exp(-\epsilon)}{(1 + \exp(-\epsilon))^2}$ | $\tanh(\epsilon/2)$ | |
| Gumbel | $\exp(-\epsilon - \exp(-\epsilon))$ | $1 - \exp(-\epsilon)$ | |
| Cauchy | $\dfrac{1}{\pi \cdot (1 + \epsilon^2)}$ | $\dfrac{2 \cdot \epsilon}{1 + \epsilon^2}$ | |
| Laplace | $\nicefrac{1}{2} \cdot \exp(-|\epsilon|)$ | $\operatorname{sign}(\epsilon)$ | |
| Triangular | $\max(0, 1 - |\epsilon|)$ | $\dfrac{\operatorname{sign}(\epsilon)}{1 - |\epsilon|}$ (∗) | |

## 2.1 DISTRIBUTION EXAMPLES

After covering the underlying theory of generalized stochastic smoothing, in this section, we provide examples of specific distributions that our theory applies to. We illustrate the distributions in Table 1.

Before delving into individual choices for distributions, we provide a clarification for multivariate densities $\mu : \mathbb{R}^n \to \mathbb{R}_{\geq 0}$: We consider the $n$-dimensional multivariate form of a distribution as the concatenation of $n$ independent univariate distributions. Thus, for $\mu_1$ as the univariate formulation of the density, we have the proportionality $\mu(\epsilon) \simeq \prod_{i=1}^{n} \mu_1(\epsilon_i)$. We remark that the distribution by which we smooth ($\mathbf{L}\epsilon$) is not an isotropic (per-dimension independent) distribution. Instead, through transformation by the scale matrix $\mathbf{L}$, e.g., in the case of the Gaussian distribution, $\mathbf{L}\epsilon$ covers the entire space of multivariate Gaussian distributions with arbitrary covariance matrices.

Beyond the **Gaussian** distribution, the **logistic** distribution offers heavier tails, and the **Gumbel** distribution provides max-stability, which can be important for specific tasks. The **Cauchy** distribution [30], with its undefined mean and infinite variance, also has important implications in smoothing: e.g., the Cauchy distribution is shown to provide monotonicity in differentiable sorting networks [12]. While prior art [22] heuristically utilized the Cauchy distribution for stochastic smoothing of argmax, this had been, thus far, without a general formal justification.

In this work, for the first time, we consider Laplace and triangular distributions. First, the **Laplace** distribution, as the symmetric extension of the exponential distribution, does not lie in the space of exponential family distributions, and is not differentiable at 0. Via Lemma 3, we show that stochastic smoothing can still be applied and is exactly correct despite non-differentiablity of the distribution. Second, with the **triangular** distribution, we illustrate, for the first time, that stochastic smoothing can be performed even with a non-differentiable distribution with compact support ($[-1, 1]$). This is crucial if the domain of $f$ has to be limited to a compact set rather than the real domain, or in applications where smoothing beyond a limited distance to the original point is not meaningful. A hypothetical application for this could be differentiating a physical motor controlled robot in reinforcement learning where we may not want to support an infinite range for safety considerations.

## 2.2 VARIANCE REDUCTION

Given an unbiased estimator of the gradient, e.g., in its simplest form (2), we desire reducing its variance, or, in other words, improve the quality of the gradient estimate for a given number of samples. For this, we consider 3 orthogonal perspectives of variance reduction: covariates, antithetic samples, and (randomized) quasi-Monte Carlo.

To illustratively derive the first two variance reductions, let us consider the case of smoothing a constant function $f(x) = v$ for some large constant $v \gg 0$. Naturally, $f_\epsilon(x) = v$ and $\nabla_x f_\epsilon(x) = 0$.

However, for a finite number of samples $s$, our empirical estimate (e.g. (7)) will differ from 0 almost surely. As the gradient of $f_\epsilon(x) - c$ wrt. $x$ does not depend on $c$, we have $\nabla_x f_\epsilon(x) = \nabla_x (f_\epsilon(x) - c)$. If we choose $c = v$, the variance of the gradient estimator is reduced to 0. For general and non-constant $f$, we can estimate the optimal choice of $c$ via $c = f(x)$ or via the leave-one-out estimator [31], [32] of $f_\epsilon(x)$. In the fields of stochastic smoothing of optimizers and reinforcement learning this is known as the *method of covariates*. $f(x)$ and LOO were previously considered for smoothing, e.g., in [22] and [33], respectively. We illustrate the effects of both choices of covariates in Figure 1.

From an orthogonal perspective, we observe that $\mathbb{E}_{\epsilon \sim \mu}[\nabla_\epsilon - \log \mu(\epsilon)] = 0$, which follows, e.g., from $\nabla_x f_\epsilon(x) = 0 = \mathbb{E}_{\epsilon \sim \mu}[v \cdot \nabla_\epsilon - \log \mu(\epsilon)]$. For symmetric distributions, we can guarantee an empirical estimate to be 0 by always using pairs of *antithetic samples* [34], i.e., complementary $\epsilon$s. Using $\epsilon' = -\epsilon$, we have $\nabla_\epsilon \log \mu(\epsilon) + \nabla_{\epsilon'} \log \mu(\epsilon') = 0$. This is illustrated in Figure 2 (2). In our experiments in the next section, we observe antithetic sampling to generally perform poorly in comparison to other variance reduction techniques.

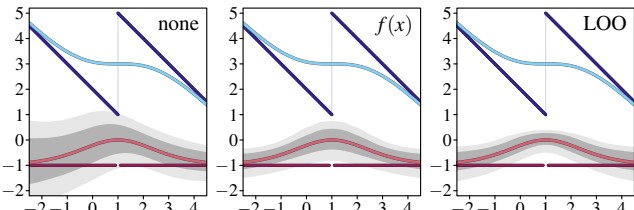

Figure 1: Comparison of covariates: a non-differentiable function (dark blue) is smoothed with a logistic distribution (light blue). The original gradient (dark red) is not everywhere defined, and does not meaningfully represent the gradient. The gradient of the smoothed function is shown in pink. Grey illustrates the variance of a gradient estimate with 5 samples via the $[25\%, 75\%]$ (dark grey) and $[10\%, 90\%]$ (light grey) percentiles. Using $f(x)$ as a covariate, instead of using *none* reduces the gradient variance, in particular whenever $f(x)$ is large. Leave-one-out (LOO) further improves over $f(x)$ at discontinuities of the original function $f$ (i.e., at $x=1$), but has slightly higher variance than $f(x)$ where $f$ is continuous and has large values (i.e., at $x=-2$.)

A third perspective considers that points sampled with standard Monte Carlo (MC) methods (see Fig. 2 (1)), due to the random nature of the sampling, often form (accidental) clumps while other areas are void of samples. To counteract this, quasi-Monte Carlo (QMC) methods [35] spread out sampled points as evenly as possible by foregoing randomness and choosing points from a regular grid, e.g., a simple Cartesian grid, taking the grid cell centers as samples (see Fig. 2 (3)). Via the inverse CDF of the respective distribution, the points can be mapped from the unit hypercube to samples from a respective distribution. However, discarding randomness makes the sampling process deterministic, limits the dispersion introduced by the smoothing distribution to concrete points, and hence makes the estimator biased. Randomized quasi-Monte Carlo (RQMC) [36] methods overcome this difficulty by reintroducing some randomness. Like QMC, RQMC uses a grid to subdivide $[0, 1]^n$ into cells, but then samples a point from each cell (see Fig. 2 (4)) instead of taking the grid cell center. While regular MC sampling leads to variances of $\mathcal{O}(1/s)$, RQMC reduces them to $\mathcal{O}(1/s^{1+2/n})$ for a number $s$ of samples and an input dimension of $n$ [37]. The default (i.e., Cartesian) QMC and RQMC methods require numbers of samples $s = k^n$ for $k \in \mathbb{N}_+$, which can become infeasible for large input dimensionalities. Thus, we also consider Latin-Hypercube Sampling (LHS) [38], which uses a subset of grid cells such that each interval in every dimension is covered exactly once (see Fig. 2 (5+6)). Finally, we remark that, to our knowledge, QMC and RQMC sampling strategies have not been considered in the field of gradient estimation.

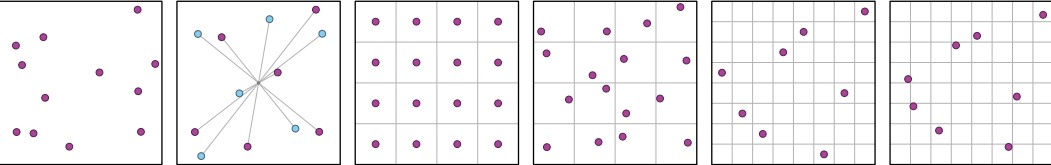

Figure 2: Sampling strategies. Left to right: Monte-Carlo (MC), Antithetic Monte-Carlo, Cartesian Quasi-Monte-Carlo (QMC), Cartesian Randomized-Quasi-Monte-Carlo (RQMC), Latin-Hypercube Sampled QMC and RQMC. Samples can be transformed via the inverse CDF of a respective distribution.

### 2.3 Smoothing of the Algorithm vs. the Objective

In algorithmic supervision settings, we can write our training objective as $\ell(h(y))$ where $y$ is the output of a neural network model, $h$ is the algorithm, and the scalar function $\ell$ is the training objective

(loss function) applied to the output of the algorithms. In such cases, we can distinguish between smoothing the algorithms ($f = h$) and smoothing the loss ($f = \ell \circ h$).

When *smoothing the algorithm*, we compute the value and derivative of $\ell\big(\mathbb{E}_\epsilon[h(y+\epsilon)]\big)$. This requires our loss function $\ell$ to be differentiable and capable of receiving relaxed inputs. (For example, if the output of $h$ is binary, then $\ell$ has to be able to operate on real-valued inputs from $(0, 1)$.) In this case, the derivative of $\mathbb{E}_\epsilon[h(y + \epsilon)]$ is a Jacobian matrix (see Corollary 2).

When *smoothing the objective / loss function*, we compute the value and derivative of $\mathbb{E}_\epsilon[\ell(h(y + \epsilon))]$. Here, the objective / loss $\ell$ does not need to be differentiable and can be limited to operate on discrete outputs of the algorithm $h$. Here, the derivative of $\mathbb{E}_\epsilon[\ell(h(y + \epsilon))]$ is a gradient.

The optimal choice between smoothing the algorithm and smoothing the objective depends on different factors including the problem setting and algorithm, the availability of a real-variate and real-valued $\ell$, and the number of samples that can be afforded. In practice, we observe that, whenever we can afford large numbers of samples, smoothing of the algorithm performs better.

## 3 RELATED WORK

In the theoretical literature of gradient-free optimization, stochastic smoothing has been extensively studied [18], [19], [39], [40]. Our work extends existing results, generalizing the set of allowed distributions, considering vector-valued functions, anisotropic scale matrices, enabling $k$-sample median differentiation, and a characterization of finite definedness of expectations and their gradients based on the relationship between characteristics of the density and smoothed functions.

From a more applied perspective, stochastic smoothing has been applied for relaxing convex optimization problems [2], [3], [22]. In particular, convex optimization formulations of argmax [2], [22], the shortest-path problem [2], and the clustering problem [3] have been considered. We remark that the perspective of smoothing any function or algorithm $f$, as in this work, differs from the perspective of perturbed optimizers. In particular, optimizers are a special case of the functions we consider.

While we consider smoothing functions with real-valued inputs, there is also a rich literature of differentiating stochastic discrete programs [41]–[43]. These works typically use the inherent stochasticity from discrete random variables in programs and explicitly model the internals of the programs. We consider real-variate black-box functions and smooth them with added input noise.

In the literature of reinforcement learning, a special case or analogous idea to stochastic smoothing can be found in the REINFORCE formulation where the (scalar) score function is smoothed via a policy [18], [20], [44], [45]. Compared to the literature, we enable new distributions and respective characterizations of requirements for the score functions. We hope our results will pave their way into future RL research directions as they are also applicable to RL without major modification.

## 4 EXPERIMENTS

For the experiments, we consider 4 experimental domains: sorting & ranking, graph algorithms, 3D mesh rendering, and cryo-electron tomography (cryoET) simulations. The primary objective of the empirical evaluations is to compare different distributions as well as different variance reduction techniques. We begin our evaluations by measuring the variance of the gradient estimators, and then continue with optimizations and using the differentiable relaxations in deep learning tasks. We remark that, in each of the 4 experiments, $f$ does not have any non-zero gradients, and thus using first-order or path-wise gradients or gradient estimators is not possible.

### 4.1 VARIANCE OF GRADIENT ESTIMATORS

We evaluate the gradient variances for different variance reduction techniques in Figures 3 and 4. For differentiable sorting and ranking, we smooth the (hard) permutation matrix that sorts an input vector ($f : \mathbb{R}^n \to \{0, 1\}^{n \times n}$). For diff. shortest-paths, we smooth the function that maps from a 2D cost-map to a binary encoding of the shortest-path under 8-neighborhood ($f : \mathbb{R}^{n \times n} \to \{0, 1\}^{n \times n}$). Both functions are not only non-differentiable, but also have no non-zero gradients anywhere. For each distribution, we compare all combinations of the 3 complementary variance reduction techniques.

On the axis of sampling strategy, we can observe that, whenever available, Cartesian RQMC delivers the lowest variance. The only exception is the triangular distribution, where latin QMC provides the lowest uncentered gradient variance (despite being a biased estimator) because of large contributions

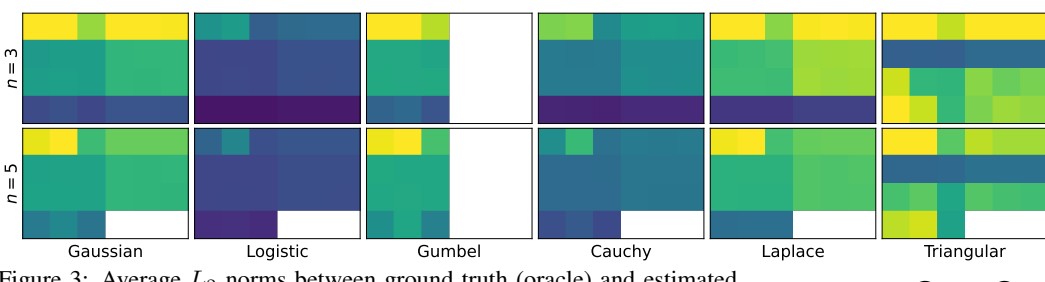

Figure 3: Average $L_2$ norms between ground truth (oracle) and estimated gradient for different numbers of elements to sort and rank $n$, and different distributions. Each plot compares different variance reduction strategies as indicated in the legend to the right of the caption. *Darker is better* (smaller values). Colors are only comparable within each subplot. We use $1\,024$ samples, except for Cartesian and $n = 3$ where we use $10^3 = 1\,000$ samples. An extension with $n \in \{7, 10\}$ can be found in Figure 11 in the appendix. Absolute values are reported in Table 8.

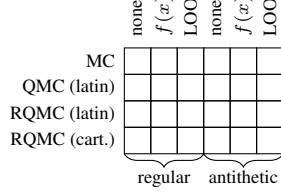

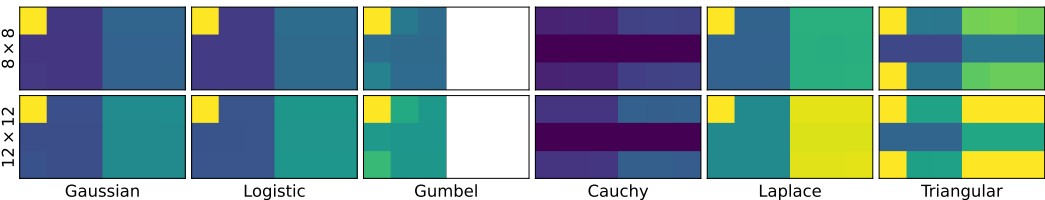

Figure 4: Average $L_2$ norms between ground truth (oracle) and estimated gradient for smoothing shortest-path algorithms, and different distributions. Each plot compares different variance reduction strategies as indicated in the legend to the right of the caption. *Darker is better* (smaller values). Colors are only comparable within each subplot. We use $1\,024$ samples. Absolute values are reported in Table 9.

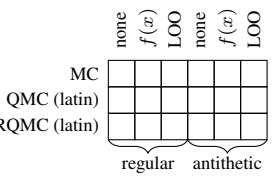

to the gradient for samples close to $-1$ and $1$. Between latin QMC and RQMC, we can observe that their variance is equal except for the high-dimension cases of the Cauchy distribution and a few cases of the Gumbel distribution, where QMC is of lower variance. However, due to the bias in QMC, RQMC would typically still be preferable over QMC. We do not consider Cartesian QMC due to its substantially greater bias. In heuristic conclusion, $\mathrm{RQMC}\,(\mathrm{c.}\,) \succ \mathrm{RQMC}\,(\mathrm{l.}\,) \succeq \mathrm{QMC}\,(\mathrm{l.}\,) \succ \mathrm{MC}$.

On the axis of using antithetic sampling (left vs. right in each subplot), we observe that it consistently performs worse than the regular counterpart, except for vanilla MC without a covariate. The reason for this is that antithetic sampling does lead to a good sample-utilization trade-off once we consider quasi Monte-Carlo strategies. For vanilla Monte-Carlo, antithetic sampling improves the results as long as we do not use the LOO covariate. Thus, in the following, we consider antithetic only for MC.

On the axis of the covariate, we observe that LOO consistently provides the lowest gradient variances. This aligns with intuition from Figure 1 where LOO provides the lowest variance at discontinuities (in this subsection, $f$ is discontinuous or constant everywhere). Comparing no covariate and $f(x)$, the better choice has a strong dependence on the individual setting, which makes sense considering the binary outputs of the algorithms. $f(x)$ would perform well for functions that attain large values while having fewer discontinuities.

In conclusion, the best setting is Cartesian RQMC with the LOO covariate and without antithetic sampling whenever available (only for $s = n^k$ samples for $k \in \mathbb{N}$). The next best choice is typically RQMC with Latin hypercube sampling.

## 4.2 DIFFERENTIABLE SORTING & RANKING

After investigating the choices of variance reduction techniques wrt. the variance alone, in this section, we explore the utility of stochastic smoothing on the 4-digit MNIST sorting benchmark [8]. Here, at each step, a set of $n=5$ 4-digit MNIST images (such as ⟦3 7 5 5⟧) is presented to a CNN, which predicts the displayed scalar value for each of the $n$ images independently. For training the model, no absolute information about the displayed value is provided, and only the ordering or ranking of the $n$ images according to their ground truth value is supervised. The goal is to learn an order-preserving

CNN, and the evaluation metric is the fraction of correctly inferred orders from the CNN (exact match accuracy). Training the CNN requires a differentiable ranking operator (that maps from a vector to a differentiable permutation matrix) for the ranking loss. Previous work has considered NeuralSort [8], SoftSort [9], casting sorting as a regularized OT problem [7], and differentiable sorting networks (DSNs) [11], [12]. The state-of-the-art is monotonic DSN [12], which utilizes a relaxation based on Cauchy distributions to provide monotonic differentiable sorting, which has strong theoretical and empirical advantages.

In Figure 5, we evaluate the performance of generalized stochastic smoothing with different distributions and different numbers of samples for each variance reduction technique. We observe that, while the Cauchy distribution performs poorly for small numbers of samples, for large numbers of samples, the Cauchy distribution performs best. This makes sense as the Cauchy distribution has infinite variance and, for DSNs, provides monotonicity. We remark that large numbers of samples can easily be afforded in many applications (when comparing the high cost of neural networks to the vanishing cost of sorting/ranking within a loss function). (Nevertheless, for $32\,768$ samples, the sorting operation starts to become the bottleneck.) The Laplace distribution is the best choice for smaller numbers of samples. Wrt. variance reduction, we continue to observe that vanilla MC performs worst. RQMC performs best, except for Triangular, where QMC is best. For the Gumbel distribution, we observe reduced performance for latin sampling. Generally, we observe that $f(x)$ is the worst choice of covariate, but the effect lies within standard deviations. In Table 2, we provide a numerical comparison to other differentiable sorting approaches. We can observe that all choices of distributions improve over all baselines except for the monotonic DSNs, even at smaller numbers of samples (i.e., without measurable impact on training speed). Finally, the Cauchy distribution leads to a small improvement over the SOTA, without requiring a manually designed differentiable sorting algorithm.

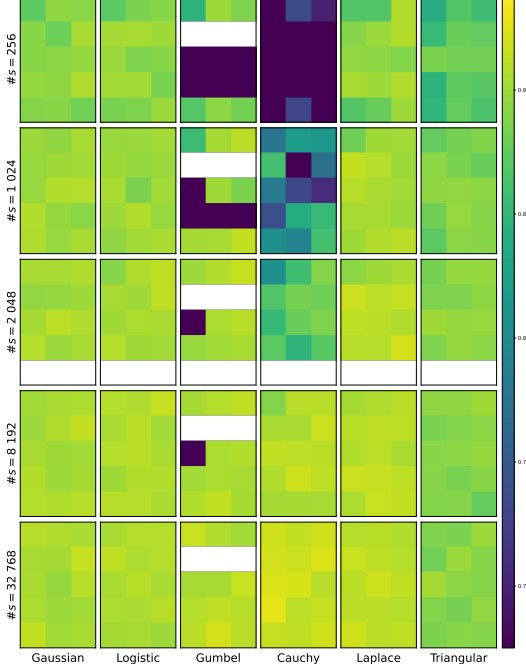

Figure 5: Sorting benchmark ($n{=}5$). Exact match (EM) accuracy. *Brighter is better* (greater values). Values between subplots are comparable. IQM over 12 seeds and displayed range of $[75\%, 85.5\%]$.

Table 2: Sorting benchmark results ($n = 5$), avg. over 12 seeds. 'best (cv)' refers to the best sampling strategy, as determined via cross-validation (thus, there is no bias from the selection of the strategy). Table 7 includes additional num. of samples and stds. Baselines are NeuralSort [8], SoftSort [9], *L*ogistic DSNs [11], *C*auchy and *E*rror-optimal DSNs [12], and OT Sort [7], avg. over at least 5 seeds each.

| Baselines | | Neu.S. | Soft.S. | L. DSN | C. DSN | E. DSN | OT. S. |
|---|---|---|---|---|---|---|---|
| — | | 71.3 | 70.7 | 77.2 | 84.9 | 85.0 | 81.1 |

| Sampling | #s | Gauss. | Logis. | Gumbel | Cauchy | Laplace | Trian. |
|---|---|---|---|---|---|---|---|
| vanilla | 256 | 82.3 | 82.8 | 79.2 | 68.1 | 82.6 | 81.3 |
| best (cv) | 256 | 83.1 | 82.7 | 81.6 | 55.6 | 83.7 | 82.7 |
| vanilla | 1k | 81.3 | 83.7 | 82.0 | 68.5 | 80.6 | 82.8 |
| best (cv) | 1k | 83.9 | 84.0 | 84.2 | 73.0 | 84.3 | 82.4 |
| vanilla | 32k | 84.2 | 84.1 | 84.5 | 84.9 | 84.4 | 83.4 |
| best (cv) | 32k | 84.4 | 84.4 | 84.8 | 85.1 | 84.4 | 84.0 |

## 4.3 DIFFERENTIABLE SHORTEST-PATHS

The Warcraft shortest-path benchmark [17] is the established benchmark for differentiable shortest-path algorithms (e.g., [2], [5], [17]). Here, a Warcraft pixel map is provided, a CNN predicts a $12 \times 12$ cost matrix, a differentiable algorithm computes the shortest-path, and the supervision is only the ground truth shortest-path. Berthet et al. [2] considered stochastic smoothing with Fenchel-Young (FY) losses, which improves sample efficiency for small numbers of samples. However, the FY loss does not improve for larger numbers of samples (e.g., Tab. 7.5 in [4]). As computing the shortest-path is computationally efficient and parallelizable (our implementation is $\approx 5\,000\times$ faster than the

Dijkstra implementation used in previous work [2], [17]), we can afford substantially larger numbers of samples, improving the quality of gradient estimation. In Figure 6, we compare the performance of different smoothing strategies. The logistic distribution performs best, and smoothing of the algorithm (top) performs better than smoothing of the loss (bottom). Variance reduction via sampling strategies (antithetic, QMC, or RQMC) improves performance, and the best covariate is LOO. For reference, the FY loss [2] leads to an accuracy of 80.6%, regardless of the number of samples. GSS consistently achieves 90%+ using 100 samples (see Fig. 13 right).

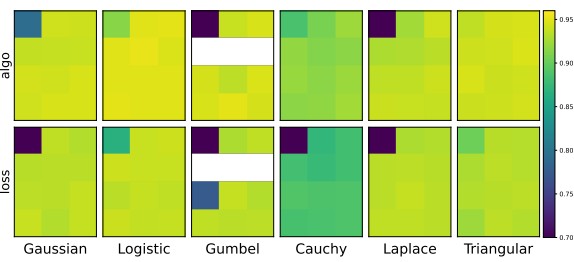

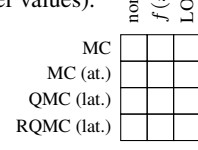

Figure 6: Warcraft shortest-path experiment with 1000 samples. *Brighter is better* (larger values). Values between subplots are comparable. Exact match accuracy avg. over 5 seeds and displayed range [70%, 96%]. Additional settings in Figures 13 and 14.

Using 10 000 samples, and variance reduction, we achieve 96.6% (Fig. 14) compared to the SOTA of 95.8% [5], a reduction of the error from 4.2% to 3.4%. In Fig. 7, we illustrate that smaller standard deviations (larger $\beta$) are better for more samples.

### 4.4 DIFFERENTIABLE RENDERING

For differentiable rendering [22], [46]–[51], we smooth a non-differentiable hard renderer via sampling. This differs from DRPO [22], which uses stochastic smoothing to relax the Heaviside and Argmax functions within an already differentiable renderer. Instead, we consider the renderer as a black-box function. This has the advantage of noise parameterized in the coordinate space rather than the image space. We benchmark stochastic smoothing for rendering by optimizing the camera-pose (4-DoF) for a Utah teapot, an experiment inspired by [22], [51]. We illustrate the results in Figure 8. Here, the logistic distribution performs best, and QMC/RQMC as well as LOO lead to the largest improvements. While Fig. 8 shows smoothing the rendering algorithm, Fig. 12 performs smoothing of the training objective / loss. Smoothing the algorithm is better because the loss (MSE), while well-defined on discrete renderings, is less meaningful on discrete renderings.

### 4.5 DIFFERENTIABLE CRYO-ELECTRON TOMOGRAPHY

Transmission Electron Microscopy (TEM) transmits electron beams through thin specimens to form images [52]. Due to the small electron beam wavelength, TEM leads to higher resolutions of up to single columns of atoms. Obtaining high resolution images from TEM involves adjustments of various experimental parameters. We apply smoothing to a realistic black-box TEM simulator [53], optimizing sets of parameters to approximate reference Tobacco Mosaic Virus (TMV) [54] micrographs. In

Figure 7: Warcraft shortest-path experiment using Gaussian smoothing of the algorithm (RQMC with latin hypercube-sampling and LOO covariate). Comparing the effects between the inverse temperature $\beta$ and the number of samples. We observe that with growing numbers of samples, the optimal inverse temperature increases, i.e., the optimal standard deviation for the Gaussian noise decreases. Averaged over 5 seeds.

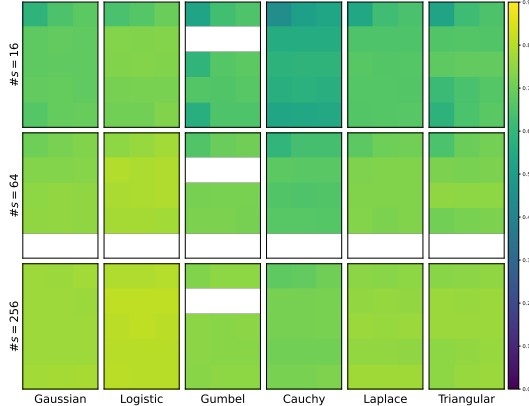

Figure 8: Utah teapot camera pose optimization. The metric is fraction of camera poses recovered; the initialization angle errors are uniformly distributed in $[15°, 75°]$. *Brighter is better.* Avg. over 768 seeds. The displayed range is $[0\%, 90\%]$.

Figure 9, we perform two experiments: a 2-parameter study optimizing the microscope acceleration voltage and $x$-position of the specimen, and a 4-parameter study with additional parameters of the particle's $y$-position and the primary lens focal length. The micrograph image sizes are $400 \times 400$ pixels, and accordingly we use smoothing of the loss.

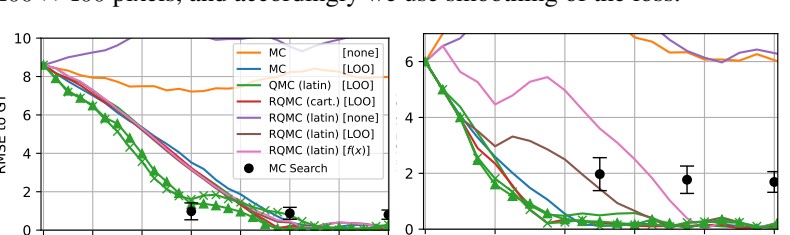

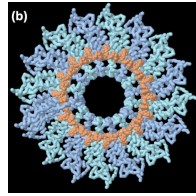

Figure 9: RMSE to Ground Truth parameters for the 2-parameter (left) and 4-parameter experiment (right). We optimize the $L_2$ loss between generated and GT images using loss smoothing. No marker lines correspond to Gaussian, $\times$ to Laplace and $\triangle$ to Triangular distributions. Laplace and Triangular perform best; LOO leads to the largest improvements. Add. results are in Figure 15.

Figure 10: (a) Simulated Transmission Electron micrograph, (b) TMV structure, with RNA (orange) and protein stacks (blue).

SUMMARY OF EXPERIMENTAL RESULTS

Generally, we observe that QMC and RQMC perform best, whereas antithetic sampling performs rather poorly. In low-dimensional problems, it is advisable to use RQMC (cartesian), and in higher dimensional problems (R)QMC (latin), still works well. As for the covariate, LOO typically performs best; however, the choice of sampling strategy (QMC/RQMC) is more important than choosing the covariate. In sorting and ranking, the Cauchy distribution performs best for large numbers of samples and for smaller numbers of samples, the Laplace distribution performs best. In the shortest-path case, the logistic distribution performs best, and Gaussian closely follows. Here, we also observe that with larger numbers of samples, the optimal standard deviation decreases. For differentiable rendering, the logistic distribution performs best.

LIMITATIONS

A limitation of our work is that zeroth-order gradient estimators are generally only competitive if the first-order gradients do not exist (see [55] for discussions on exceptions). In this vein, in order to be competitive with custom designed continuous relaxations like a differentiable renderer, we may need a very large number of samples, which could become prohibitive for expensive functions $f$. The optimal choice of distribution depends on the function to be smoothed, which means there is no singular distribution that is optimal for all $f$; however, if one wants to limit the distribution to a single choice, we recommend the logistic or Laplace distribution, as, with their simple exponential convergence, they give a good middle ground between heavy-tailed and light-tailed distributions. Finally, the variance reduction techniques like QMC/RQMC are not immediately applicable in single sample settings, and the variance reduction techniques in this paper build on evaluating $f$ many times.

## 5 CONCLUSION

In this work, we derived stochastic smoothing with reduced assumptions and outline a general framework for relaxation and gradient estimation of non-differentiable black-box functions. This enables an increased set of distributions for stochastic smoothing, e.g., enabling smoothing with the triangular distribution while maintaining full differentiablility of $f_\epsilon$. We investigated variance reduction for stochastic smoothing–based gradient estimation from 3 orthogonal perspectives, finding that RQMC and LOO are generally the best methods, whereas the popular antithetic sampling method performs rather poorly. Moreover, enabled by supporting vector-valued functions, we disentangled the algorithm and objective, thus smoothing $f$ while analytically backpropagating through the loss $\ell$, improving gradient estimation. We applied stochastic smoothing to differentiable sorting and ranking, diff. shortest-paths on graphs, diff. rendering for pose estimation and diff. cryo-ET simulations. We hope that our work inspires the community to develop their own stochastic relaxations for differentiating non-differentiable algorithms, operators, and simulators.

REPRODUCIBILITY STATEMENT

We provide proofs for all theoretical statements. For the statements in the main paper, we provide proofs in Appendix A. We provide experimental details in Appendix E including specification of hyperparameters, a list of assets, and runtimes for our experiments. Alongside this submission, we upload the source code of generalized stochastic smoothing. At publication, we will publicly release the code.

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

# A  PROOFS

## A.1  PROOF OF LEMMA 3

*Proof of Lemma 3.* Let $\Omega \subset \mathbb{R}^n$ be the set of values where $\nabla_\epsilon \mu(\epsilon)$ is undefined. $\mu$ is differentiable a.e. and $\Omega$ has Lebesgue measure $0$.

Recapitulating (5) from the proof of Lemma 1, we have

$$\nabla_x f_\epsilon(x) = - \int f(x + \epsilon) \, \nabla_\epsilon \mu(\epsilon) \, d\epsilon \,. \tag{14}$$

Replacing $\nabla_\epsilon \mu(\epsilon)$ by *any* of the weak derivatives $\nu$ of $\mu$, which exists and is integrable due to absolute continuity, we have

$$\nabla_x f_\epsilon(x) = - \int f(x + \epsilon) \, \nu(\epsilon) \, d\epsilon \tag{15}$$

$$= - \int_{\mathbb{R}^n \backslash \Omega} f(x + \epsilon) \, \nu(\epsilon) \, d\epsilon - \int_\Omega f(x + \epsilon) \, \nu(\epsilon) \, d\epsilon \,. \tag{16}$$

Because $\mu$ is absolutely continuous and as the Lebesgue measure of $\Omega$ is $0$, per Hölder's inequality

$$\int_\Omega |f(x + \epsilon) \, \nu(\epsilon)| \, d\epsilon \leq \int_\Omega |f(x + \epsilon)| \, d\epsilon \cdot \int_\Omega |\nu(\epsilon)| \, d\epsilon = \int_\Omega |f(x + \epsilon)| \, d\epsilon \cdot 0 = 0 \tag{17}$$

where $\int_\Omega |\nu(\epsilon)| \, d\epsilon = 0$ follows from absolute continuity of $\mu$. Thus,

$$\int_\Omega f(x + \epsilon) \, \nu(\epsilon) \, d\epsilon = 0 \,. \tag{18}$$

As $\nu = \nabla_\epsilon \mu(\epsilon)$ for all $\epsilon \in \mathbb{R}^n \backslash \Omega$

$$\nabla_x f_\epsilon(x) = - \int_{\mathbb{R}^n \backslash \Omega} f(x + \epsilon) \, \nu(\epsilon) \, d\epsilon - \int_\Omega f(x + \epsilon) \, \nu(\epsilon) \, d\epsilon = - \int_{\mathbb{R}^n \backslash \Omega} f(x + \epsilon) \, \nabla_\epsilon \mu(\epsilon) \, d\epsilon \,, \tag{19}$$

showing that for all possible choices of $\nu$, the gradient estimator coincides. Thus, we complete our proof via

$$\nabla_x f_\epsilon(x) = - \int_{\mathbb{R}^n \backslash \Omega} f(x + \epsilon) \, \mu(\epsilon) \, \nabla_\epsilon \log \mu(\epsilon) \, d\epsilon = \mathbb{E}_{\epsilon \sim \mu} \left[ f(x + \epsilon) \cdot \mathbf{1}_{\epsilon \notin \Omega} \cdot \nabla_\epsilon - \log \mu(\epsilon) \right] \,. \tag{20}$$

After completing the proof, we remark that, if the density was not continuous, e.g., uniform $\mathcal{U}([0, 1])$, then $\int_{\{0\}} \nabla_\epsilon \mu(\epsilon) \, d\epsilon = \left[ \mu(\epsilon) \right]_{\epsilon \nearrow 0}^{\epsilon \searrow 0} = 1$. This means that the weak derivative is not defined (or loosely speaking "the derivative is infinity"), thereby violating the assumptions of Hölder's inequality (Eq. 17). This concludes that continuity is required for the proof to hold. $\qquad \square$

## A.2 Proof of Lemma 6

*Proof of Lemma 6.*

$$
\begin{align}
\nabla_\gamma f_{\gamma\epsilon}(x) &= \nabla_\gamma \mathbb{E}_{\epsilon\sim\mu}\left[f(x+\gamma\cdot\epsilon)\right] \tag{21} \\
&= \nabla_\gamma \int f(x+\gamma\cdot\epsilon)\mu(\epsilon)d\epsilon \tag{22} \\
&\quad \left(u = x + \epsilon\cdot\gamma \Rightarrow \epsilon = \tfrac{u-x}{\gamma} \; ; \; \tfrac{du}{d\epsilon} = \gamma \Rightarrow d\epsilon = \tfrac{1}{\gamma}du\right) \tag{23} \\
&= \nabla_\gamma \int f(u)\mu(\epsilon)\tfrac{1}{\gamma}du \tag{24} \\
&= \int f(u)\nabla_\gamma(\mu(\epsilon)\tfrac{1}{\gamma})du \tag{25} \\
&= \int f(u)(\tfrac{1}{\gamma}\nabla_\gamma\mu(\epsilon) + \mu(\epsilon)\nabla_\gamma\tfrac{1}{\gamma})du \tag{26} \\
&= \int f(u)(\tfrac{1}{\gamma}(\nabla_\epsilon\mu(\epsilon))^\top\tfrac{\partial\epsilon}{\partial\gamma} - \mu(\epsilon)\tfrac{1}{\gamma^2})du \tag{27} \\
&= \int f(u)(\tfrac{1}{\gamma}(\nabla_\epsilon\mu(\epsilon))^\top\tfrac{\partial}{\partial\gamma}\tfrac{u-x}{\gamma} - \mu(\epsilon)\tfrac{1}{\gamma^2})du \tag{28} \\
&= \int f(u)(\tfrac{1}{\gamma}(\nabla_\epsilon\mu(\epsilon))^\top(-\tfrac{\epsilon}{\gamma}) - \tfrac{1}{\gamma^2}\mu(\epsilon))du \tag{29} \\
&= \int f(u)(-(\nabla_\epsilon\mu(\epsilon))^\top\epsilon - \mu(\epsilon))\tfrac{1}{\gamma^2}du \tag{30} \\
&\quad \left(\nabla_\epsilon\log\mu(\epsilon) = \tfrac{1}{\mu(\epsilon)}\nabla_\epsilon\mu(\epsilon) \Rightarrow \nabla_\epsilon\mu(\epsilon) = \mu(\epsilon)\nabla_\epsilon\log\mu(\epsilon)\right) \tag{31} \\
&= \int f(u)(-(\mu(\epsilon)\nabla_\epsilon\log\mu(\epsilon))^\top\epsilon - \mu(\epsilon))\tfrac{1}{\gamma^2}\cdot\underbrace{\gamma\,d\epsilon}_{=du} \tag{32} \\
&= \int f(u)\cdot(-(\nabla_\epsilon\log\mu(\epsilon))^\top\epsilon - 1)\cdot\tfrac{1}{\gamma}\cdot\mu(\epsilon)\,d\epsilon \tag{33} \\
&= \int f(u)\cdot(-1 + (\nabla_\epsilon-\log\mu(\epsilon))^\top\epsilon)\cdot\tfrac{1}{\gamma}\cdot\mu(\epsilon)\,d\epsilon \tag{34} \\
&= \mathbb{E}_{\epsilon\sim\mu}\left[f(x+\gamma\cdot\epsilon)\cdot\left(-1 + (\nabla_\epsilon-\log\mu(\epsilon))^\top\cdot\epsilon\right)/\gamma\right]. \tag{35}
\end{align}
$$

$\square$

## A.3 Proof of Theorem 7

*Proof of Theorem 7.*

**Part 1:** $\partial f_{\mathbf{L}\epsilon}(x)\,/\,\partial x$

We perform a change of variables, $u = x + \mathbf{L}\epsilon \implies \epsilon = \mathbf{L}^{-1}(u-x)$ and

$$
d\epsilon = \frac{du}{du}d\epsilon = \frac{d\epsilon}{du}du = \frac{d\mathbf{L}^{-1}(u-x)}{du}du = \frac{d\mathbf{L}^{-1}u}{du}du = \det(\mathbf{L}^{-1})\,du \tag{36}
$$

Thus,

$$
f_{\mathbf{L}\epsilon}(x) = \int f(x+\mathbf{L}\epsilon)\mu(\epsilon)\,d\epsilon = \int f(u)\cdot\mu(\mathbf{L}^{-1}(u-x))\cdot\det(\mathbf{L}^{-1})\,du. \tag{37}
$$

Now,

$$\nabla_x f_{\mathbf{L}\epsilon}(x)_i = \nabla_x \int f(u)_i \cdot \mu(\mathbf{L}^{-1}(u-x)) \cdot \det(\mathbf{L}^{-1}) \, du \tag{38}$$

$$= \int f(u)_i \cdot \nabla_x \big( \mu(\mathbf{L}^{-1}(u-x)) \big) \cdot \det(\mathbf{L}^{-1}) \, du \tag{39}$$

$$= \int f(u)_i \cdot \mathbf{L}^{-1} \cdot \big( \nabla_\epsilon - \mu(\epsilon) \big) \cdot \det(\mathbf{L}^{-1}) \, du \tag{40}$$

$$= \int f(x+\mathbf{L}\epsilon)_i \cdot \mathbf{L}^{-1} \cdot \nabla_\epsilon - \mu(\epsilon) \, d\epsilon \tag{41}$$

$$= \int f(x+\mathbf{L}\epsilon)_i \cdot \mathbf{L}^{-1} \cdot \mu(\epsilon) \cdot \nabla_\epsilon - \log \mu(\epsilon) \, d\epsilon \tag{42}$$

$$= \mathbb{E}_{\epsilon \sim \mu} \Big[ f(x+\mathbf{L}\epsilon)_i \cdot \mathbf{L}^{-1} \cdot \nabla_\epsilon - \log \mu(\epsilon) \Big] \tag{43}$$

**Part 2:** $\partial f_{\mathbf{L}\epsilon}(x) \, / \, \partial \mathbf{L}$

We use the same change of variables as above.

$$\nabla_{\mathbf{L}} \, \mathbb{E}_{\epsilon \sim \mu} \big[ f(x+\mathbf{L} \cdot \epsilon)_i \big] \tag{44}$$

$$= \nabla_{\mathbf{L}} \int f(x+\mathbf{L}\epsilon)_i \cdot \mu(\epsilon) \, d\epsilon \tag{45}$$

$$= \nabla_{\mathbf{L}} \int f(u)_i \cdot \mu(\mathbf{L}^{-1}(u-x)) \cdot \det(\mathbf{L}^{-1}) \, du \tag{46}$$

$$= \int f(u)_i \cdot \nabla_{\mathbf{L}} \Big( \mu(\mathbf{L}^{-1}(u-x)) \cdot \det(\mathbf{L}^{-1}) \Big) \, du \tag{47}$$

$$= \int f(x+\mathbf{L}\epsilon)_i \cdot \nabla_{\mathbf{L}} \Big( \mu(\mathbf{L}^{-1}(u-x)) \cdot \det(\mathbf{L}^{-1}) \Big) / \det(\mathbf{L}^{-1}) \, d\epsilon \tag{48}$$

$$= \mathbb{E}_{\epsilon \sim \mu} \Big[ f(x+\mathbf{L}\epsilon)_i \cdot \nabla_{\mathbf{L}} \Big( \mu(\mathbf{L}^{-1}(u-x)) \cdot \det(\mathbf{L}^{-1}) \Big) \cdot \det(\mathbf{L}) \, / \mu(\epsilon) \Big] \tag{49}$$

Now, while $\nabla_{\mathbf{L}} \Big( \mu(\mathbf{L}^{-1}(u-x)) \cdot \det(\mathbf{L}^{-1}) \Big)$ may be computed via automatic differentiation, we can also solve it in closed-form. Firstly, we can observe that

$$\nabla_{\mathbf{L}} \mu(\mathbf{L}^{-1}(u-x)) = \nabla_{\epsilon^\top} \mu(\epsilon) \cdot \nabla_{\mathbf{L}} (\mathbf{L}^{-1} \cdot (u-x)) \tag{50}$$

$$= \nabla_{\mathbf{L}} (\nabla_{\epsilon^\top} \mu(\epsilon) \cdot \mathbf{L}^{-1} \cdot (u-x)) \tag{51}$$

$$= -\mathbf{L}^{-\top} \cdot \nabla_\epsilon \mu(\epsilon) \cdot (u-x)^\top \cdot \mathbf{L}^{-\top} \tag{52}$$

and

$$\nabla_{\mathbf{L}} \det(\mathbf{L}^{-1}) = -\det(\mathbf{L})^{-1} \cdot \mathbf{L}^{-\top} . \tag{53}$$

We can combine this to resolve it in closed form to:

$$\nabla_{\mathbf{L}} \Big( \mu(\mathbf{L}^{-1}(u-x)) \cdot \det(\mathbf{L}^{-1}) \Big) = -\mathbf{L}^{-\top} \cdot \nabla_\epsilon \mu(\epsilon) \cdot (u-x)^\top \cdot \mathbf{L}^{-\top} \cdot \det(\mathbf{L}^{-1})$$
$$- \mu(\mathbf{L}^{-1}(u-x)) \cdot \det(\mathbf{L})^{-1} \cdot \mathbf{L}^{-\top} \tag{54}$$

$$= -\mathbf{L}^{-\top} \cdot \nabla_\epsilon \mu(\epsilon) \cdot \big( \mathbf{L}^{-1}(u-x) \big)^\top \cdot \det(\mathbf{L}^{-1})$$
$$- \mu(\epsilon) \cdot \det(\mathbf{L})^{-1} \cdot \mathbf{L}^{-\top} \tag{55}$$

$$= -\mathbf{L}^{-\top} \cdot \nabla_\epsilon \mu(\epsilon) \cdot \epsilon^\top \cdot \det(\mathbf{L}^{-1})$$
$$- \mu(\epsilon) \cdot \det(\mathbf{L})^{-1} \cdot \mathbf{L}^{-\top} \tag{56}$$

$$= -\det(\mathbf{L}^{-1}) \cdot \big( \mathbf{L}^{-\top} \cdot \nabla_\epsilon \mu(\epsilon) \cdot \epsilon^\top + \mu(\epsilon) \cdot \mathbf{L}^{-\top} \big) . \tag{57}$$

Combing this with equation (49), we have

$$\nabla_{\mathbf{L}} \, \mathbb{E}_{\epsilon \sim \mu} \big[ f(x + \mathbf{L} \cdot \epsilon)_i \big]$$

$$= \mathbb{E}_{\epsilon \sim \mu} \left[ f(x + \mathbf{L}\epsilon)_i \cdot \nabla_{\mathbf{L}} \Big( \mu(\mathbf{L}^{-1}(u - x)) \cdot \det(\mathbf{L^{-1}}) \Big) \cdot \det(\mathbf{L}) \, / \, \mu(\epsilon) \right]$$

$$= \mathbb{E}_{\epsilon \sim \mu} \left[ f(x + \mathbf{L}\epsilon)_i \, \cdot \, -\det(\mathbf{L^{-1}}) \cdot \big(\mathbf{L}^{-\top} \cdot \nabla_\epsilon \mu(\epsilon) \cdot \epsilon^\top + \mu(\epsilon) \cdot \mathbf{L}^{-\top}\big) \, \cdot \, \det(\mathbf{L}) \, / \, \mu(\epsilon) \right] \quad (58)$$

$$= \mathbb{E}_{\epsilon \sim \mu} \left[ f(x + \mathbf{L}\epsilon)_i \, \cdot \, -\big(\mathbf{L}^{-\top} \cdot \nabla_\epsilon \mu(\epsilon) \cdot \epsilon^\top + \mu(\epsilon) \cdot \mathbf{L}^{-\top}\big) \, / \, \mu(\epsilon) \right] \quad (59)$$

$$= \mathbb{E}_{\epsilon \sim \mu} \left[ f(x + \mathbf{L}\epsilon)_i \, \cdot \, -\big(\mathbf{L}^{-\top} \cdot \nabla_\epsilon \mu(\epsilon) \cdot \epsilon^\top \, / \, \mu(\epsilon) + \mathbf{L}^{-\top}\big) \right] \quad (60)$$

$$= \mathbb{E}_{\epsilon \sim \mu} \left[ f(x + \mathbf{L}\epsilon)_i \, \cdot \, \mathbf{L}^{-\top} \, \cdot \, \big( -1 + \nabla_\epsilon - \log \mu(\epsilon) \cdot \epsilon^\top \big) \right] . \quad (61)$$

$\square$

# B   DISCUSSION OF PROPERTIES OF $f$ FOR FINITELY DEFINED $f_\epsilon$ AND $\nabla f_\epsilon$

When we have a function $f$ that is not defined with a compact range with $f : \mathbb{R}^n \to \mathbb{R}$, and have a density $\mu$ with unbounded support (e.g., Gaussian or Cauchy), we may experience $f_\epsilon$ or even $\nabla f_\epsilon$ to not be finitely defined. For example, virtually any distribution with full support on $\mathbb{R}$ leads to the smoothing $f_\epsilon$ of the degenerate function $f : x \mapsto \exp(\exp(\exp(\exp(x^2))))$ to not be finitely defined.

We say a function, as described via an expectation, is finitely defined iff it is defined (i.e., the expectation has a value) and its value is finite (i.e., not infinity). For example, the first moment of the Cauchy distribution is undefined, and the second moment is infinite; thus, both moments are not finitely defined.

We remark that the considerations in this appendix also apply to prior works that enable the real plane as the output space of $f$. We further remark that writing an expression for smoothing and the gradient of a arbitrary function with non-compact range is not necessarily false; however, e.g., any claim that smoothness is guaranteed if the gradient jumps from $-\infty$ to $\infty$ (e.g., the power tower in the first paragraph) is not formally correct. We remark that characterizing valid $f$s via a Lipschitz or other continuity requirement is not applicable because this would defeat the goal of differentiating non-differentiable and discontinuous $f$.

In the following, we discuss when $f_\epsilon$ or $\nabla f_\epsilon$ are finitely defined. For this, let us cover a few preliminaries:

Let a function $f(x)$ be called $\mathcal{O}(b(x))$ bounded if there exist $c, v \in \mathcal{O}(b(x))$ and $\bar{c}, \bar{v} \in \mathbb{R}$ such that

$$\bar{c} + c(x) \le f(x) \le \bar{v} + v(x) \qquad \forall x . \quad (62)$$

For example, a function may be called polynomially bounded (wrt. a polynomial $b(x)$) if (but not only if) $-b(x) \le f(x) \le b(x)$.

Moreover, let a density $\mu$ with support $\mathbb{R}$ be called decaying faster than $b(x)$ if $\mu(x) \in o(b(x))$. For example, the standard Gaussian density decays faster than $\exp(-|x|)$, i.e., $\mu(x) \in o(\exp(-|x|))$. Additionally, we can say that Gaussian density decays at rate $\exp(-x^2)$, i.e., $\mu(x) \in \theta(\exp(-x^2))$.

Now, we can formally characterize finite definedness of $f_\epsilon$ and $\nabla f_\epsilon$:

**Lemma 8** (Finite Definedness of $f_\epsilon$). *$f_\epsilon$ is finitely defined if there exists an increasing function $b(\cdot)$ such that*

$$f(x) \text{ is bounded by } \mathcal{O}(b(x)) \qquad \text{and} \qquad \mu(\epsilon) \in \mathcal{O}(1/b(\epsilon + \alpha\epsilon)/\epsilon^{(1+\alpha)}) \quad (63)$$

*for some $\alpha > 0$.*

*Proof.* To show that $f_\epsilon$ exists, we need to show that

$$\int_{\mathbb{R}} \big| f(x + \epsilon) \cdot \mu(\epsilon) \big| \, d\epsilon \quad (64)$$

is finite for all $x$. Let $\tilde{f}$ be an absolutely upper bound of $f$, and w.l.o.g. let us choose $\tilde{f}(y) = b(y) + \bar{b}$ with $b(y) > 1$ for $y \in \mathbb{R}$. Further, as per the assumptions $\mu(\epsilon) < \frac{1}{\epsilon^{(1+\alpha)} \cdot b(\epsilon + \alpha\epsilon)} \cdot w$ for all $\epsilon < \omega_1$ as well as all $\epsilon > \omega_2$ for some $w, \omega_1, \omega_2$. Let us restrict $\omega_1, \omega_2$ to $\omega_1 < -|x|/\alpha$ and $\omega_2 > |x|/\alpha$. It is trivial to see that

$$\int_{\omega_1}^{\omega_2} \left| f(x + \epsilon) \cdot \mu(\epsilon) \right| d\epsilon < \infty. \tag{65}$$

W.l.o.g., let us consider the upper remainder:

$$\int_{\omega_2}^{\infty} \left| f(x + \epsilon) \cdot \mu(\epsilon) \right| d\epsilon \leq \int_{\omega_2}^{\infty} \left| \tilde{f}(x + \epsilon) \cdot \mu(\epsilon) \right| d\epsilon \tag{66}$$

$$\leq \int_{\omega_2}^{\infty} \left| (b(x + \epsilon) + \bar{b}) \cdot \frac{1}{\epsilon^{(1+\alpha)} \cdot b(\epsilon + \alpha\epsilon)} \cdot w \right| d\epsilon \tag{67}$$

$$= \int_{\omega_2}^{\infty} \left| \left( \frac{b(x + \epsilon)}{\epsilon^{(1+\alpha)} \cdot b(\epsilon + \alpha\epsilon)} + \frac{\bar{b}}{\epsilon^{(1+\alpha)} \cdot b(\epsilon + \alpha\epsilon)} \right) \cdot w \right| d\epsilon \tag{68}$$

$$\leq \int_{\omega_2}^{\infty} \left| \left( \frac{b(x + \epsilon)}{\epsilon^{(1+\alpha)} \cdot b(\epsilon + |x|)} + \frac{\bar{b}}{\epsilon^{(1+\alpha)} \cdot b(\epsilon + \alpha\epsilon)} \right) \cdot w \right| d\epsilon \tag{69}$$

$$\leq \int_{\omega_2}^{\infty} \left| \left( \frac{1}{\epsilon^{(1+\alpha)}} + \frac{\bar{b}}{\epsilon^{(1+\alpha)} \cdot b(\epsilon + \alpha\epsilon)} \right) \cdot w \right| d\epsilon \tag{70}$$

$$< \int_{\omega_2}^{\infty} \left| \frac{1}{\epsilon^{(1+\alpha)}} + \frac{\bar{b}}{\epsilon^{(1+\alpha)}} \right| d\epsilon \cdot w \tag{71}$$

$$= \int_{\omega_2}^{\infty} \left| \frac{1}{\epsilon^{(1+\alpha)}} \right| d\epsilon \cdot w \cdot (1 + \bar{b}) < \infty. \tag{72}$$

That $\int_{\omega_2}^{\infty} \frac{1}{\epsilon^{(1+\alpha)}} d\epsilon$ is finite for the step in (72) can be shown via

$$\int_{\omega_2}^{\infty} \frac{1}{\epsilon^{(1+\alpha)}} d\epsilon = \int_{\omega_2}^{\infty} \epsilon^{-1-\alpha} d\epsilon = \left[ -\frac{1}{\alpha} \epsilon^{-\alpha} \right]_{\omega_2}^{\infty} = \left[ -\frac{1}{\alpha} \lim_{\epsilon \to \infty} \epsilon^{-\alpha} + \frac{1}{\alpha} \omega_2^{-\alpha} \right] = \frac{1}{\alpha} \omega_2^{-\alpha}.$$

The same can be shown analogously for the integral $\int_{-\infty}^{\omega_1}$. This completes the proof. $\square$

**Lemma 9** (Finite Definedness of $\nabla f_\epsilon$). $\nabla f_\epsilon$ *is finitely defined if there exists an increasing function* $b(\cdot)$ *such that*

$$f(x) \text{ is bounded by } \mathcal{O}(b(x)) \qquad and \qquad \left| \mu(\epsilon) \cdot \nabla_\epsilon - \log \mu(\epsilon) \right| \in \mathcal{O}(1/b(\epsilon + \alpha\epsilon)/\epsilon^{(1+\alpha)}) \tag{73}$$

*for some $\alpha > 0$.*

*Proof.* The proof of Lemma 8 also applies here, but with $\left| \mu(\epsilon) \cdot \nabla_\epsilon - \log \mu(\epsilon) \right| < \frac{1}{\epsilon^{(1+\alpha)} \cdot b(\epsilon + \alpha\epsilon)} \cdot w$ for all $\epsilon < \omega_1$ as well as all $\epsilon > \omega_2$ for some $w, \omega_1, \omega_2$. $\square$

**Example 10** (Cauchy and the Identity). Let $\mu$ be the density of a Cauchy distribution and let $f(x) = x$. The tightest $b$ for $f(x) \in \mathcal{O}(b(x))$ is $b(x) = x$.

We have $\mu(\epsilon) \in \theta(1/\epsilon^2)$ and thus $\mu(\epsilon) \notin o(1/\epsilon^2)$. $f_\epsilon$, i.e., the mean of the Cauchy distribution is not defined.

However, its gradient $\nabla f_\epsilon = 1$ is indeed finitely defined. In particular, we can see that

$$\mu(\epsilon) \cdot \nabla_\epsilon - \log \mu(\epsilon) = \frac{2\epsilon}{\pi \cdot (1 + \epsilon^2) \cdot (1 + \epsilon^2)} \in \theta(1/\epsilon^3). \tag{74}$$

This is an intriguing property of the Cauchy distribution (or other edge cases) where $f_\epsilon$ is undefined whereas $\nabla f_\epsilon$ is finitely and well-defined. In practice, we often only require the gradient for stochastic gradient descent, which means that we often only require $\nabla f_\epsilon$ to be well defined and do not necessarily need to evaluate $f_\epsilon$ depending on the application.

Additional discussions for the Cauchy distribution and an extension of stochastic smoothing to the $k$-sample median can be found in the next appendix.

## C  Stochastic Smoothing, Medians, and the Cauchy Distribution

In this section, we provide a discussion of a special case of stochastic smoothing with the Cauchy distribution, and provide an extension of stochastic smoothing to the $k$-sample median. This becomes important if the range of $f$ is not subset of a compact set, and thus $\mathbb{E}_{\epsilon \sim \mu}\big[f(x+\epsilon)\big]$ becomes undefined for some choice of distribution $\mu$. For example, for $f(x+\epsilon) = \epsilon$ and $\mu$ being the density of a Cauchy distribution, $\mathbb{E}_{\epsilon \sim \mu}\big[f(x+\epsilon)\big] = \mathbb{E}_{\epsilon \sim \mu}\big[\epsilon\big]$ is undefined. Nevertheless, even in this case, the gradient estimators discussed in this paper for $\nabla_x \mathbb{E}_{\epsilon \sim \mu}\big[f(x+\epsilon)\big]$ remain well defined. This is practically relevant because $\mathbb{E}_{\epsilon \sim \mu}\big[f(x+\epsilon)\big]$ does not need to be finitely defined as long as $\nabla_x \mathbb{E}_{\epsilon \sim \mu}\big[f(x+\epsilon)\big]$ is well defined. Further, we remark that the undefinedness of $\mathbb{E}_{\epsilon \sim \mu}\big[f(x+\epsilon)\big]$ requires the range of $f$ to be unbounded, i.e., if there exists a maximum / minimum possible output, then it is well defined. Moreover, there exist $f$ with unbounded range for which $\mathbb{E}_{\epsilon \sim \mu}\big[f(x+\epsilon)\big]$ also remains well defined.

To account for cases where $\mathbb{E}_{\epsilon \sim \mu}\big[f(x+\epsilon)\big]$ may not be well defined or not a robust statistic, we introduce an extension of smoothing to the median. We begin by defining the $k$-sample median.

**Definition 11** ($k$-Sample Median). For a number of samples $k > 1$, and a distribution $\zeta$, we say that

$$\mathbb{E}_{z_1, z_2, \ldots, z_k \sim \zeta}\Big[ \text{median}\, \{z_1, z_2, \ldots, z_k\} \Big] \tag{75}$$

is the $k$-sample median. For multivariate distributions, let $\text{median}$ be the per-dimension median.

Indeed, for $k \geq 5$, the $k$-sample median estimator is shown to have finite variance for the Cauchy distribution (Theorem 3 and Example 2 in [56]), which implies a well defined $k$-sample median. Moreover, for any distribution with a density of the median bounded away from 0, the first and second moments are guaranteed to be finitely defined for sufficiently large $k$. This is important for non-trivial $f$ with $f(\epsilon) \neq \epsilon$ for at least one $\epsilon$ with $\epsilon \sim \mu$, which implies $\zeta \neq \mu$. Thus, rather than computing and differentiating the expected value, we can differentiate the $k$-sample median.

**Lemma 12** (Differentiation of the $k$-Sample Median). *With the $k$-sample median smoothing as*

$$f_\epsilon^{(k)}(x) = \mathbb{E}_{\epsilon_1, \ldots, \epsilon_k \sim \mu}\Big[ \text{median}\, \{f(x+\epsilon_1), \ldots, f(x+\epsilon_k)\} \Big], \tag{76}$$

*we can differentiate $f_\epsilon^{(k)}(x)$ as*

$$\nabla_x f_\epsilon^{(k)}(x) = \mathbb{E}_{\epsilon_1, \ldots, \epsilon_k \sim \mu}\Big[ f(x+\epsilon_{r(\epsilon)}) \cdot \nabla_{\epsilon_{r(\epsilon)}} - \log \mu(\epsilon_{r(\epsilon)}) \Big] \tag{77}$$

*where $r(\epsilon)$ is the arg-median of the set $\{f(x+\epsilon_1), \ldots, f(x+\epsilon_k)\}$, which is equivalent to the implicit definition via $f(x+\epsilon_{r(\epsilon)}) = \text{median}\, \{f(x+\epsilon_1), \ldots, f(x+\epsilon_k)\}$.*

*Proof.* We denote $\epsilon_{1:k} \sim \mu^{(1:k)}$ such that $\epsilon_{1:k} = \big[\epsilon_1^\top, \ldots, \epsilon_k^\top\big]^\top$ and $\epsilon_i \sim \mu \; \forall i \in \{1, \ldots, k\}$.

$$\nabla_x f_\epsilon^{(k)}(x) = \nabla_x \mathbb{E}_{\epsilon_1, \ldots, \epsilon_k \sim \mu}\Big[ \text{median}\, \{f(x+\epsilon_1), \ldots, f(x+\epsilon_k)\} \Big] \tag{78}$$

$$= \nabla_x \mathbb{E}_{\epsilon_{1:k} \sim \mu^{(1:k)}}\Big[ \text{median}\, \{f(x+\epsilon_1), \ldots, f(x+\epsilon_k)\} \Big] \tag{79}$$

$$= \nabla_x \int_{\mathbb{R}^{n \cdot k}} \text{median}\, \{f(x+\epsilon_1), \ldots, f(x+\epsilon_k)\} \cdot \mu^{(1:k)}(\epsilon_{1:k})\, d\epsilon_{1:k} \tag{80}$$

$$(x_1, \ldots, x_k = x) \quad = \sum_{j=1}^{k} \nabla_{x_j} \int_{\mathbb{R}^{n \cdot k}} \text{median}\, \{f(x_1+\epsilon_1), \ldots, f(x_k+\epsilon_k)\} \cdot \mu^{(1:k)}(\epsilon_{1:k})\, d\epsilon_{1:k} \tag{81}$$

As a shorthand, we abbreviate the indicator $\mathbb{1}_{f(x_j+\epsilon_j)=\text{median}\{f(x_1+\epsilon_1),...,f(x_k+\epsilon_k)\}}$ as $\mathbb{1}_{j,\epsilon_{1:k}}$ and abbreviate $\mathbb{1}_{f(u_j)=\text{median}\{f(u_1),...,f(u_k)\}}$ as $\mathbb{1}_{j,u_{1:k}}$:

$$\nabla_x f_\epsilon^{(k)}(x) = \sum_{j=1}^k \nabla_{x_j} \int_{\mathbb{R}^{n\cdot k}} f(x_j+\epsilon_j) \cdot \mathbb{1}_{j,\epsilon_{1:k}} \cdot \mu^{(1:k)}(\epsilon_{1:k}) \, d\epsilon_{1:k} \tag{82}$$

$$= \sum_{j=1}^k \nabla_{x_j} \int_{\mathbb{R}^{n\cdot k}} f(u) \cdot \mathbb{1}_{j,u1:k} \cdot \mu^{(1:k)}(u_{1:k}-x) \, du_{1:k} \tag{83}$$

$$= \sum_{j=1}^k \int_{\mathbb{R}^{n\cdot k}} f(u) \cdot \mathbb{1}_{j,u1:k} \cdot \nabla_{x_j} \mu^{(1:k)}(u_{1:k}-x) \, du_{1:k} \tag{84}$$

$$= \sum_{j=1}^k \int_{\mathbb{R}^{n\cdot k}} f(x+\epsilon_j) \cdot \mathbb{1}_{j,\epsilon_{1:k}} \cdot -\nabla_{\epsilon_j} \mu^{(1:k)}(\epsilon_{1:k}) \, d\epsilon_{1:k} \tag{85}$$

We have

$$\nabla_{\epsilon_j} \mu^{(1:k)}(\epsilon_{1:k}) = \mu^{(1:k)}(\epsilon_{1:k}) \cdot \nabla_{\epsilon_j} \log \mu^{(1:k)}(\epsilon_{1:k}) = \mu^{(1:k)}(\epsilon_{1:k}) \cdot \nabla_{\epsilon_j} \log \mu(\epsilon_j). \tag{86}$$

Thus,

$$\nabla_x f_\epsilon^{(k)}(x) = \sum_{j=1}^k \int_{\mathbb{R}^{n\cdot k}} f(x+\epsilon_j) \cdot \mathbb{1}_{j,\epsilon_{1:k}} \cdot -\mu^{(1:k)}(\epsilon_{1:k}) \cdot \nabla_{\epsilon_j} \log \mu(\epsilon_j) \, d\epsilon_{1:k} \tag{87}$$

$$= \int_{\mathbb{R}^{n\cdot k}} \sum_{j=1}^k \left[ \mathbb{1}_{j,\epsilon_{1:k}} \cdot f(x+\epsilon_j) \cdot \nabla_{\epsilon_j} - \log \mu(\epsilon_j) \right] \cdot \mu^{(1:k)}(\epsilon_{1:k}) \, d\epsilon_{1:k} \tag{88}$$

Indicating the choice of median in dependence of $\epsilon_{1:k}$, we define $r(\epsilon_{1:k})$ s.t. $\mathbb{1}_{r(\epsilon_{1:k}),\epsilon_{1:k}} = 1$. Thus,

$$\nabla_x f_\epsilon^{(k)}(x) = \int_{\mathbb{R}^{n\cdot k}} f(x+\epsilon_{r(\epsilon_{1:k})}) \cdot \nabla_{\epsilon_{r(\epsilon_{1:k})}} - \log \mu(\epsilon_{r(\epsilon_{1:k})}) \cdot \mu^{(1:k)}(\epsilon_{1:k}) \, d\epsilon_{1:k} \tag{89}$$

$$= \mathbb{E}_{\epsilon_{1:k}\sim\mu^{(1:k)}} \left[ f(x+\epsilon_{r(\epsilon_{1:k})}) \cdot \nabla_{\epsilon_{r(\epsilon_{1:k})}} - \log \mu(\epsilon_{r(\epsilon_{1:k})}) \right] \tag{90}$$

This concludes the proof. $\qquad\square$

Empirically, we can estimate $\nabla_x f_\epsilon^{(k)}(x)$ for $s$ propagated samples ($s > k$) without bias as

$$\nabla_x f_\epsilon^{(k)}(x) \triangleq \sum_{i=1}^s \left[ q_i \cdot f(x+\epsilon_i) \cdot \nabla_{\epsilon_i} - \log \mu(\epsilon_i) \right] \qquad \epsilon_1, ..., \epsilon_s \sim \mu \tag{91}$$

where $q_i$ is the probability of $f(x+\epsilon_i)$ being the median in a subset of $k$ samples, i.e., under uniqueness of $g_i$s, we have

$$q_i = \frac{\displaystyle\sum_{\{h_1,...,h_k\}\subset\{g_1,...,g_s\}} \mathbb{1}\big(g_i = \text{median}\{h_1,...,h_k\}\big)}{\displaystyle\binom{s}{k}} \qquad g_i := f(x+\epsilon_i). \tag{92}$$

We remark that, in case of non-uniqueness, it is adequate to split the probability among the candidates; however, under non-discreteness assumptions on $f$ (density of $\zeta < \infty$, the converse typically implies the range of $f$ being a subset of a compact set), this almost surely (with probability 1) does not occur.

We have shown that the $k$-sample median $f_\epsilon^{(k)}(x)$ is differentiable and demonstrated an unbiased gradient estimator for it. A straightforward extension for the case of $f$ being differentiable is differentiating through the median via a $k \to \infty$-sample median, e.g., via setting $s = k^2$. The $k \to \infty$ extension for differentiating through the median itself requires $f$ being differentiable because, for discontinuous $f$, $f_\epsilon^{(k)}(x)$ is differentiable only for $k < \infty$. (As an illustration, the median of the Heaviside function under a symmetric perturbation $\mu$ with density at $0$ bounded away from $0$ is the exactly the Heaviside function.)

## D  VARIANCES AND LIPSCHITZ CONSTANTS FOR EACH DISTRIBUTION

In this section, we provide variances of the gradient estimator $\nabla f_\epsilon$ and Lipschitz constants of $f_\epsilon$ for each of the 6 distributions considered in the paper.

Table 3: Lipschitz constants for each distribution for functions $f : R^n \to \{0, 1\}^m$.

| Distribution | Lipschitz constant |
|---|---|
| Gaussian | $\frac{1}{\sqrt{2\pi}\gamma} \approx 0.399/\gamma$ |
| Logistic | $0.25/\gamma$ |
| Gumbel | $\frac{1}{e\gamma} \approx 0.368/\gamma$ |
| Cauchy | $\frac{1}{\pi\gamma} \approx 0.318/\gamma$ |
| Laplace | $0.5/\gamma$ |
| Triangular | $1.0/\gamma$ |

In Table 4, we can see that the Laplace distribution performs best, consistently achieving a variance of 0, which however, in this particular case is due to the simple nature of the problem. The second best distribution is the logistic distribution. The logistic behavior is similar to the Laplace distribution also from an analytical perspective (the logistic can be seen as a smoothed variant of the Laplace).

Table 4: Variances in the case of the sign function ($f(x) = \text{sign}(x)$), smoothed with each respective distribution, and evaluating the variance at point 0, in dependence of the number of samples $s$. To standardize the distributions, we consider 2 settings: scaling each input distribution to a variance of 1, as well as choosing a scale such that the Lipschitz constant of the resulting smooth function is 1.

| $f(x) = \text{sign}(x)$ | Var $= 1$ | Lipschitz $= 1$ |
|---|---|---|
| Gaussian | $0.364/s$ | $0.573/s$ |
| Logistic | $0.272/s$ | $0.336/s$ |
| Gumbel | $0.759/s$ | $0.859/s$ |
| Cauchy | n/a | $0.234/s$ |
| Laplace | $0.000/s$ | $0.000/s$ |
| Triangular | $1.747/s$ | $2.496/s$ |

In the more complex setting of Table 5, we observe similar behavior. The Laplace dist. has a variance different from 0, but still has the smallest variance. Extending the variances to RQMC (latin) in Table 6, we observe similar behavior, but substantially smaller variances.

Table 5: Variances in the case of $f$ being the sign of the sine function ($f(x) = \text{sign}(\sin(x))$), smoothed with each respective distribution, and evaluating the variance at point 0, in dependence of the number of samples $s$.

| $f(x) = \text{sign}(\sin(x))$ | Var $= 1$ | Lipschitz $= 1$ |
|---|---|---|
| Gaussian | $0.382/s$ | $0.588/s$ |
| Logistic | $0.318/s$ | $0.690/s$ |
| Gumbel | $0.840/s$ | $1.130/s$ |
| Cauchy | n/a | $0.452/s$ |
| Laplace | $0.093/s$ | $0.144/s$ |
| Triangular | $1.727/s$ | $2.609/s$ |

Table 6: Variances for $f(x) = \text{sign}(x)$) with RQMC (latin) at $s = 100$ samples, which drastically reduces the variance further. As the rate is faster than $1/s$, we report it for 100 samples.

| $f(x) = \text{sign}(x)$ | RQMC, Var $= 1$ | RQMC, $L = 1$ |
|---|---|---|
| Gaussian | 0.0000224 | 0.0000355 |
| Logistic | 0.0000011 | 0.0000013 |
| Gumbel | 0.0001780 | 0.0002008 |
| Cauchy | n/a | 0.0000041 |
| Laplace | 0.0000000 | 0.0000000 |
| Triangular | 0.0109100 | 0.0156400 |

# E  EXPERIMENTAL DETAILS

**MNIST Sorting Benchmark Experiments**    We train for $100\,000$ steps at a learning rate of 0.001 with the Adam optimizer using a batch size of 100. Following the requirements of the benchmark, we use the same model as previous works [7], [8], [11]. That is, two convolutional layers with a kernel size of $5 \times 5$, 32 and 64 channels respectively, each followed by a ReLU and MaxPool layer; after flattening, this is followed by a fully connected layer with a size of 64, a ReLU layer, and a fully connected output layer mapping to a scalar. For each distribution and number of samples, we choose the optimal $\gamma \in \{1, 1/3, 0.1\}$.

**Warcraft Shortest-Path Benchmark Experiments**    Following the established protocol [17], we train for 50 epochs with the Adam optimizer at a batch size of 70 and an initial learning rate of 0.001. The learning rate decays by a factor of 10 after 30 and 40 epochs each. The model is the first block of ResNet18. The hyperparameter $\gamma = 1/\beta$ as specified in Figures 13 and 14.

**Utah Teapot Camera Pose Optimization Experiments**    We initialize the pose to be perturbed by angles uniformly sampled from $[15°, 75°]$. The ground truth orientation is randomly sampled from the sphere of possible orientations. The ground truth camera angle is $20°$, and the ground truth camera distance is uniformly sampled from $[2.5, 4]$. The initial camera distance is sampled as being uniformly offset by $[-0.5, 6]$, thus the feasible set of initial camera distance guesses lies in $[2, 10]$. The initial camera angle is uniformly sampled from $[10°, 30°]$. We optimize for $1\,000$ steps with the Adam optimizer $[(\beta_1, \beta_2) = (0.5, 0.99)]$ and the CosineAnnealingLR scheduler with an initial learning rate of 0.3. We schedule the diagonal of $\mathbf{L}$ to decay exponentially from $[0.1, 5°, 5°, 0.25°] \cdot 10^{0.75}$ to $[0.1, 5°, 5°, 0.25°] \cdot 10^{-1.75}$ (the dimensions are camera distance, 2 pose angles, and the camera angle). As discussed, the success criterion is finding the angle within $5°$ of the ground truth angle. There is typically no local minimum within $5°$ and it is a reliable indicator for successful alignment.

**Differentiable Cryo-Electron Tomography Experiments**    The ground truth values of the parameters are set to 300 kV for acceleration voltage, 3 mm for the focal length, and the ground truth sample specimen is centered as $(x, y) = (0, 0)$ nm units. For reporting errors, the acceleration voltages are normalized by a factor of 100 to ensure that all parameters vary over commensurate ranges. For the 2-parameter optimization, the feasible set of acceleration voltage varied over a range of $[0, 1000]$ kV and the feasible set of the specimen's $x$-position varied over the range $[-5, 5]$. For the 4-parameter optimization, the feasible set of acceleration voltage varied over a range of $[0, 600]$ kV, the focal length ranges over $[0, 6]$ mm, the $x$- and $y$-positions range over $[-3, 3]$. We use the Adam optimizer for both experiments, with $[(\beta_1, \beta_2) = (0.5, 0.9)]$. For the MC Search baseline, we generate sets of $n$ uniform random points in the feasible region of the parameters, generate micrographs for these random parameter tuples using the TEM simulator [53], and identify the parameter tuple in the set having the lowest mean squared error with respect to the ground truth image. The RMSE between this parameter tuple and the ground truth parameters is the metric for the specific set of $n$ randomly generated values. This is repeated 20 times to obtain the mean and standard deviation of the RMSE metric at that $n$.

## E.1  ASSETS

*List of assets:*

- The sixth platonic solid (aka. Teapotahedron or Utah tea pot) [57]   [License N/A]
- Multi-digit MNIST [8], which builds on MNIST [58]   [MIT License / CC License]
- Warcraft shortest-path data set [17]   [MIT License]
- PyTorch [59]   [BSD 3-Clause License]
- TEM-simulator [53]   [GNU General Public License]

## E.2  RUNTIMES

The runtimes for sorting and shortest-path experiments are for one full training on 1 GPU. The pose optimization experiment runtimes are the total time for all 768 seeds on 1 GPU. For the TEM-

simulator, we report the CPU time per simulation sample, which is the dominant and only the measureable component of the total optimization routine time. The choice of distribution, covariate, and choice of variance reduction does not have a measurable effect on training times.

- MNIST Sorting Benchmark Experiments [1 Nvidia V100 GPU]
  - Training w/ 256 samples:        65 min
  - Training w/ 1 024 samples:       67 min
  - Training w/ 2 048 samples:       68 min
  - Training w/ 8 192 samples:       77 min
  - Training w/ 32 768 samples:   118 min
- Warcraft Shortest-Path Benchmark Experiments [1 Nvidia V100 GPU]
  - Training w/ 10 samples:          9 min
  - Training w/ 100 samples:        19 min
  - Training w/ 1 000 samples:       26 min
  - Training w/ 10 000 samples:   101 min
- Utah Teapot Camera Pose Optimization Experiments [1 Nvidia A6000 GPU]
  - Optimization on 768 seeds w/ 16 samples: 25 min
  - Optimization on 768 seeds w/ 64 samples: 81 min
  - Optimization on 768 seeds w/ 256 samples: 362 min
- Differentiable Cryo-Electron Tomography Experiments [CPU: 44 Intel Xeon Gold 5118]
  - Simulator time per sample on 1 CPU core: 67 sec

# F    ADDITIONAL EXPERIMENTAL RESULTS

Table 7: Extension of Table 2 with additional numbers of samples and standard deviations.

| Baselines | | Neu.S. | Soft.S. | L. DSN | C. DSN | E. DSN | OT. S. |
|---|---|---|---|---|---|---|---|
| — | | 71.3 | 70.7 | 77.2 | 84.9 | 85.0 | 81.1 |
| Sampling | #s | Gauss. | Logis. | Gumbel | Cauchy | Laplace | Trian. |
| vanilla | 256 | 82.3±2.0 | 82.8±0.9 | 79.2±9.7 | 68.1±19.3 | 82.6±0.8 | 81.3±1.2 |
| best (cv) | 256 | 83.1±1.6 | 82.7±1.8 | 81.6±3.6 | 55.6±13.3 | 83.7±0.8 | 82.7±1.1 |
| vanilla | 1024 | 81.3±9.1 | 83.7±0.7 | 82.0±1.6 | 68.5±24.8 | 80.6±9.0 | 82.8±1.0 |
| best (cv) | 1024 | 83.9±0.6 | 84.0±0.5 | 84.2±0.6 | 73.0±12.6 | 84.3±0.6 | 82.4±1.6 |
| vanilla | 2048 | 84.1±0.6 | 83.6±0.8 | 84.0±0.5 | 75.7±11.6 | 83.8±0.7 | 83.2±0.6 |
| best (cv) | 2048 | 84.2±0.5 | 84.2±0.6 | 84.6±0.4 | 82.0±2.2 | 84.8±0.5 | 83.4±0.5 |
| vanilla | 8192 | 84.0±0.6 | 84.2±0.8 | 84.0±0.6 | 83.6±1.0 | 83.9±1.0 | 83.6±0.7 |
| best (cv) | 8192 | 84.4±0.6 | 84.5±0.5 | 84.1±0.7 | 84.3±0.5 | 84.3±0.4 | 83.7±0.4 |
| vanilla | 32768 | 84.2±0.5 | 84.1±0.4 | 84.5±0.7 | 84.9±0.5 | 84.4±0.5 | 83.4±0.8 |
| best (cv) | 32768 | 84.4±0.4 | 84.4±0.4 | 84.8±0.5 | 85.1±0.4 | 84.4±0.4 | 84.0±0.3 |

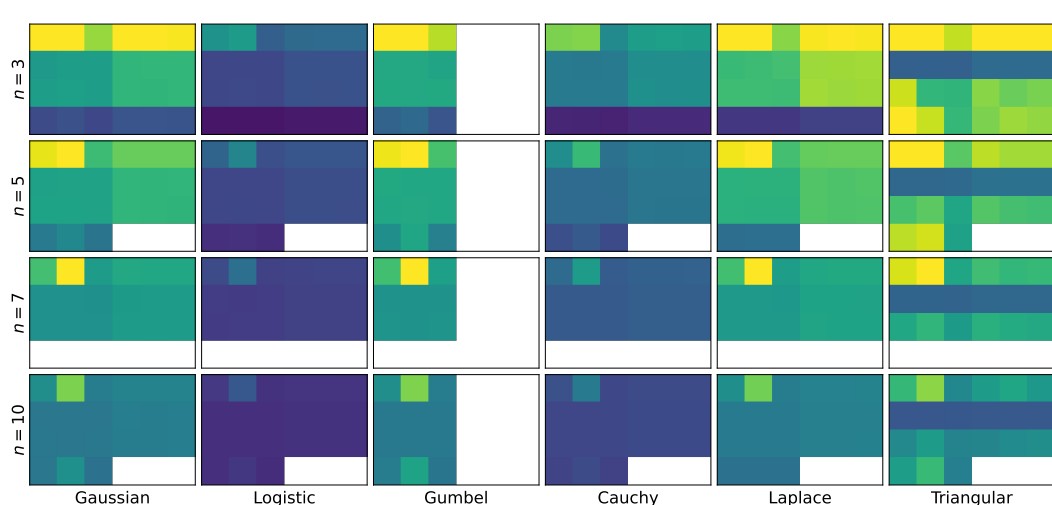

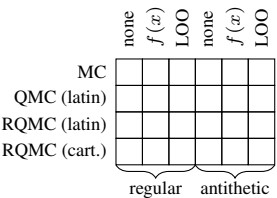

Figure 11: Average $L_2$ norms between ground truth (oracle) and estimated gradient for different numbers of elements to sort and rank $n$, and different distributions. Each plot compares different variance reduction strategies as indicated in the legend to the right of the caption. *Darker is better* (smaller values). Colors are only comparable within each subplot. We use $1\,024$ samples, except for Cartesian and $n = 3$ where we use $10^3 = 1\,000$ samples.

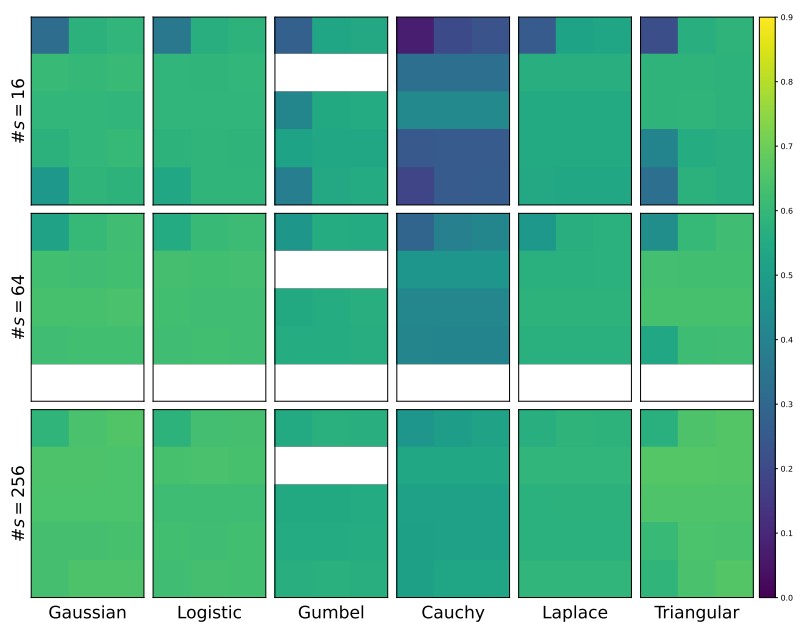

Figure 12: Utah teapot camera pose optimization with *smoothing of the loss*, compared to Figure 8, which performs *smoothing of the algorithm*. Smoothing the algorithm is consistently better, with the largest effect for larger numbers of samples. Results averaged over 768 seeds.

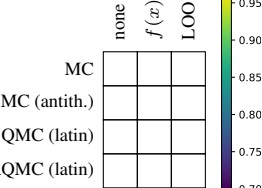

Figure 13: Warcraft shortest-path experiment. *Left: 10 samples. Right: 100 samples.* Averaged over 5 seeds. *Brighter is better.* Values between subplots are comparable. The displayed range is $[70\%, 96.5\%]$.

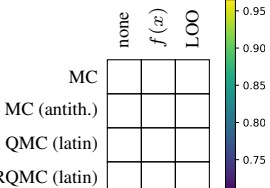

Figure 14: Warcraft shortest-path experiment. *Left: 1 000 samples. Right: 10 000 samples.* Averaged over 5 seeds. *Brighter is better.* Values between subplots are comparable. The displayed range is $[70\%, 96.5\%]$.

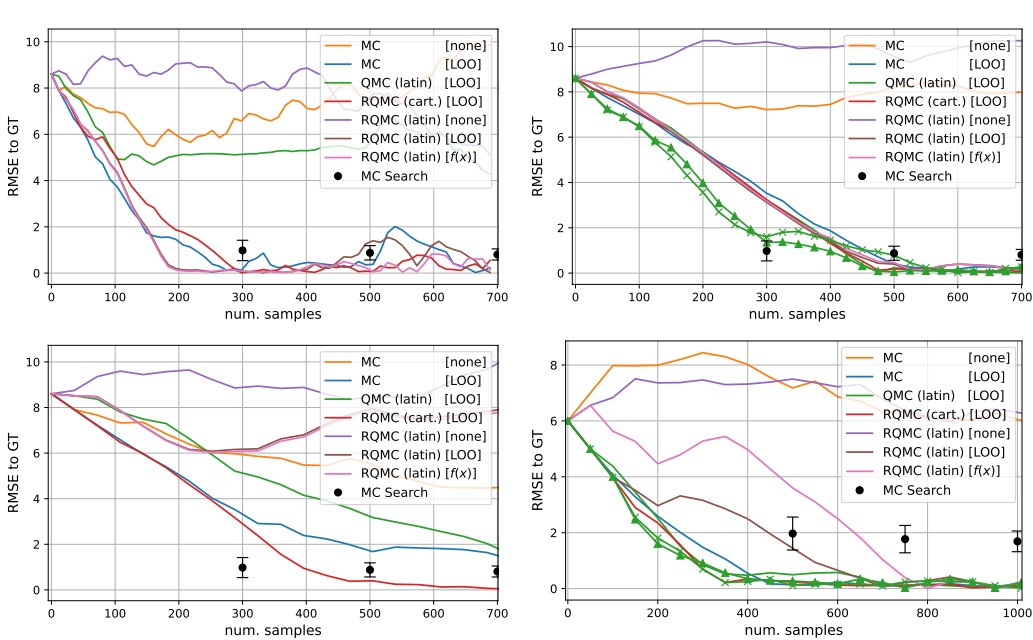

Figure 15: Cryo-Electron Tomography Experiments: RMSE with respect to Ground Truth parameters for different number of parameters optimized and for different number of samples per optimization step: (Top Left) 2-parameters & number of samples=9, (Top Right) 2-parameters & number of samples=25, (Bottom Left) 2-parameters & number of samples=36, (Bottom Right) 4-parameters. No marker lines correspond to Gaussian, × corresponds to Laplace, and △ corresponds to Triangular distributions. Ascertaining optimal parameters with minimal evaluations is important not just for high resolution imaging, but also to minimize radiation damage to the specimen. In this light, of the covariate choices, LOO generally leads to best improvement and *none* consistently leads to deterioration in performance. The Laplace and Triangular distributions lead to best performance. For the Gaussian distribution, Cartesian RQMC is generally exhibiting best results.

Table 8: Individual absolute values from the variance simulations for differentiable sorting in Figure 3. The minimum and values within 1% of the minimum are indicated as bold.

(a) values for Gaussian $(n = 3)$

|  | none | $f(x)$ | LOO | none | $f(x)$ | LOO |
|---|---|---|---|---|---|---|
|  | regular | | | antithetic | | |
| MC | 0.0084 | 0.0079 | 0.0046 | 0.0055 | 0.0054 | 0.0053 |
| QMC (lat.) | 0.0029 | 0.0030 | 0.0030 | 0.0036 | 0.0036 | 0.0036 |
| RQMC (l.) | 0.0030 | 0.0030 | 0.0030 | 0.0036 | 0.0035 | 0.0036 |
| RQMC (c.) | 0.0012 | 0.0013 | **0.0012** | 0.0014 | 0.0014 | 0.0014 |

(b) values for Gaussian $(n = 5)$

|  | none | $f(x)$ | LOO | none | $f(x)$ | LOO |
|---|---|---|---|---|---|---|
|  | regular | | | antithetic | | |
| MC | 0.0241 | 0.0308 | 0.0171 | 0.0192 | 0.0192 | 0.0192 |
| QMC (lat.) | 0.0143 | 0.0144 | 0.0144 | 0.0164 | 0.0164 | 0.0164 |
| RQMC (l.) | 0.0145 | 0.0145 | 0.0144 | 0.0164 | 0.0164 | 0.0162 |
| RQMC (c.) | 0.0103 | 0.0116 | **0.0097** | — | — | — |

(c) values for Logistic $(n = 3)$

|  | none | $f(x)$ | LOO | none | $f(x)$ | LOO |
|---|---|---|---|---|---|---|
|  | regular | | | antithetic | | |
| MC | 0.0028 | 0.0030 | 0.0016 | 0.0019 | 0.0019 | 0.0019 |
| QMC (lat.) | 0.0012 | 0.0012 | 0.0012 | 0.0014 | 0.0014 | 0.0014 |
| RQMC (l.) | 0.0012 | 0.0012 | 0.0012 | 0.0014 | 0.0013 | 0.0014 |
| RQMC (c.) | **0.0003** | 0.0003 | **0.0003** | 0.0004 | 0.0004 | 0.0004 |

(d) values for Logistic $(n = 5)$

|  | none | $f(x)$ | LOO | none | $f(x)$ | LOO |
|---|---|---|---|---|---|---|
|  | regular | | | antithetic | | |
| MC | 0.0081 | 0.0114 | 0.0061 | 0.0067 | 0.0067 | 0.0067 |
| QMC (lat.) | 0.0053 | 0.0053 | 0.0054 | 0.0060 | 0.0060 | 0.0060 |
| RQMC (l.) | 0.0053 | 0.0054 | 0.0053 | 0.0060 | 0.0060 | 0.0059 |
| RQMC (c.) | 0.0033 | 0.0036 | **0.0033** | — | — | — |

(e) values for Gumbel $(n = 3)$

|  | none | $f(x)$ | LOO | none | $f(x)$ | LOO |
|---|---|---|---|---|---|---|
|  | regular | | | antithetic | | |
| MC | 0.0086 | 0.0082 | 0.0048 | — | — | — |
| QMC (lat.) | 0.0033 | 0.0033 | 0.0032 | — | — | — |
| RQMC (l.) | 0.0033 | 0.0033 | 0.0033 | — | — | — |
| RQMC (c.) | 0.0017 | 0.0018 | **0.0014** | — | — | — |

(f) values for Gumbel $(n = 5)$

|  | none | $f(x)$ | LOO | none | $f(x)$ | LOO |
|---|---|---|---|---|---|---|
|  | regular | | | antithetic | | |
| MC | 0.0243 | 0.0323 | 0.0177 | — | — | — |
| QMC (lat.) | 0.0151 | 0.0149 | 0.0150 | — | — | — |
| RQMC (l.) | 0.0150 | 0.0151 | 0.0150 | — | — | — |
| RQMC (c.) | 0.0124 | 0.0148 | **0.0109** | — | — | — |

(g) values for Cauchy $(n = 3)$

|  | none | $f(x)$ | LOO | none | $f(x)$ | LOO |
|---|---|---|---|---|---|---|
|  | regular | | | antithetic | | |
| MC | 0.0043 | 0.0044 | 0.0026 | 0.0030 | 0.0030 | 0.0030 |
| QMC (lat.) | 0.0022 | 0.0022 | 0.0022 | 0.0027 | 0.0027 | 0.0027 |
| RQMC (l.) | 0.0022 | 0.0022 | 0.0022 | 0.0027 | 0.0026 | 0.0027 |
| RQMC (c.) | 0.0006 | 0.0006 | **0.0005** | 0.0006 | 0.0006 | 0.0006 |

(h) values for Cauchy $(n = 5)$

|  | none | $f(x)$ | LOO | none | $f(x)$ | LOO |
|---|---|---|---|---|---|---|
|  | regular | | | antithetic | | |
| MC | 0.0123 | 0.0169 | 0.0094 | 0.0102 | 0.0101 | 0.0102 |
| QMC (lat.) | 0.0088 | 0.0087 | 0.0088 | 0.0098 | 0.0098 | 0.0098 |
| RQMC (l.) | 0.0088 | 0.0088 | 0.0087 | 0.0098 | 0.0097 | 0.0097 |
| RQMC (c.) | 0.0061 | 0.0070 | **0.0056** | — | — | — |

(i) values for Laplace $(n = 3)$

|  | none | $f(x)$ | LOO | none | $f(x)$ | LOO |
|---|---|---|---|---|---|---|
|  | regular | | | antithetic | | |
| MC | 0.0086 | 0.0074 | 0.0044 | 0.0054 | 0.0054 | 0.0054 |
| QMC (lat.) | 0.0037 | 0.0037 | 0.0038 | 0.0046 | 0.0046 | 0.0047 |
| RQMC (l.) | 0.0037 | 0.0037 | 0.0037 | 0.0047 | 0.0046 | 0.0046 |
| RQMC (c.) | **0.0009** | **0.0009** | **0.0009** | 0.0010 | 0.0011 | 0.0010 |

(j) values for Laplace $(n = 5)$

|  | none | $f(x)$ | LOO | none | $f(x)$ | LOO |
|---|---|---|---|---|---|---|
|  | regular | | | antithetic | | |
| MC | 0.0245 | 0.0305 | 0.0176 | 0.0191 | 0.0192 | 0.0192 |
| QMC (lat.) | 0.0159 | 0.0160 | 0.0160 | 0.0182 | 0.0180 | 0.0182 |
| RQMC (l.) | 0.0160 | 0.0159 | 0.0159 | 0.0182 | 0.0181 | 0.0181 |
| RQMC (c.) | **0.0091** | **0.0091** | **0.0091** | — | — | — |

(k) values for Triangular $(n = 3)$

|  | none | $f(x)$ | LOO | none | $f(x)$ | LOO |
|---|---|---|---|---|---|---|
|  | regular | | | antithetic | | |
| MC | 0.1191 | 0.0683 | 0.0490 | 0.0659 | 0.0624 | 0.0602 |
| QMC (lat.) | **0.0166** | 0.0169 | **0.0166** | 0.0189 | 0.0188 | 0.0188 |
| RQMC (l.) | 0.0498 | 0.0358 | 0.0352 | 0.0444 | 0.0417 | 0.0431 |
| RQMC (c.) | 0.0682 | 0.0494 | 0.0361 | 0.0435 | 0.0461 | 0.0452 |

(l) values for Triangular $(n = 5)$

|  | none | $f(x)$ | LOO | none | $f(x)$ | LOO |
|---|---|---|---|---|---|---|
|  | regular | | | antithetic | | |
| MC | 0.3329 | 0.2779 | 0.1857 | 0.2255 | 0.2157 | 0.2149 |
| QMC (lat.) | **0.0844** | **0.0845** | **0.0851** | 0.0932 | 0.0931 | 0.0928 |
| RQMC (l.) | 0.1768 | 0.1872 | 0.1479 | 0.1827 | 0.1765 | 0.1737 |
| RQMC (c.) | 0.2251 | 0.2325 | 0.1430 | — | — | — |

Table 9: Individual absolute values from the variance simulations for differentiable shortest-paths in Figure 4. The minimum and values within 1% of the minimum are indicated as bold.

(a) values for Gaussian $(8 \times 8)$

|  | none | $f(x)$ | LOO | none | $f(x)$ | LOO |
|---|---|---|---|---|---|---|
|  | regular | | | antithetic | | |
| MC | 1330.01 | 4.17 | 4.17 | 8.32 | 8.32 | 8.34 |
| QMC (lat.) | **4.04** | **4.04** | **4.04** | 8.04 | 8.04 | 8.07 |
| RQMC (l.) | 4.25 | **4.05** | **4.05** | 8.10 | 8.09 | 8.12 |

(b) values for Gaussian $(12 \times 12)$

|  | none | $f(x)$ | LOO | none | $f(x)$ | LOO |
|---|---|---|---|---|---|---|
|  | regular | | | antithetic | | |
| MC | 6800.98 | 20.93 | 20.95 | 41.82 | 41.78 | 41.88 |
| QMC (lat.) | **20.60** | **20.60** | **20.65** | 41.12 | 41.11 | 41.18 |
| RQMC (l.) | 21.69 | **20.66** | **20.68** | 41.31 | 41.33 | 41.42 |

(c) values for Logistic $(8 \times 8)$

|  | none | $f(x)$ | LOO | none | $f(x)$ | LOO |
|---|---|---|---|---|---|---|
|  | regular | | | antithetic | | |
| MC | 1449.44 | 4.53 | 4.53 | 9.04 | 9.04 | 9.05 |
| QMC (lat.) | **4.42** | **4.42** | **4.43** | 8.80 | 8.80 | 8.83 |
| RQMC (l.) | **4.44** | **4.44** | **4.44** | 8.88 | 8.87 | 8.90 |

(d) values for Logistic $(12 \times 12)$

|  | none | $f(x)$ | LOO | none | $f(x)$ | LOO |
|---|---|---|---|---|---|---|
|  | regular | | | antithetic | | |
| MC | 7447.38 | 22.83 | 22.86 | 45.62 | 45.61 | 45.75 |
| QMC (lat.) | **22.56** | **22.56** | **22.61** | 45.01 | 44.99 | 45.07 |
| RQMC (l.) | **22.66** | **22.65** | **22.68** | 45.30 | 45.32 | 45.41 |

(e) values for Gumbel $(8 \times 8)$

|  | none | $f(x)$ | LOO | none | $f(x)$ | LOO |
|---|---|---|---|---|---|---|
|  | regular | | | antithetic | | |
| MC | 2275.31 | 10.35 | 9.08 | — | — | — |
| QMC (lat.) | 9.11 | **8.84** | **8.85** | — | — | — |
| RQMC (l.) | 11.33 | **8.91** | **8.91** | — | — | — |

(f) values for Gumbel $(12 \times 12)$

|  | none | $f(x)$ | LOO | none | $f(x)$ | LOO |
|---|---|---|---|---|---|---|
|  | regular | | | antithetic | | |
| MC | 11642.74 | 52.89 | 46.11 | — | — | — |
| QMC (lat.) | 46.88 | **45.41** | **45.48** | — | — | — |
| RQMC (l.) | 58.12 | **45.74** | **45.80** | — | — | — |

(g) values for Cauchy $(8 \times 8)$

|  | none | $f(x)$ | LOO | none | $f(x)$ | LOO |
|---|---|---|---|---|---|---|
|  | regular | | | antithetic | | |
| MC | 249027.67 | 263426.66 | 255440.59 | 507004.19 | 525973.88 | 509764.25 |
| QMC (lat.) | **2533.24** | **2532.93** | **2537.32** | **2531.24** | **2532.92** | **2537.35** |
| RQMC (l.) | 251018.28 | 267124.91 | 264146.84 | 476293.00 | 507766.00 | 529030.06 |

(h) values for Cauchy $(12 \times 12)$

|  | none | $f(x)$ | LOO | none | $f(x)$ | LOO |
|---|---|---|---|---|---|---|
|  | regular | | | antithetic | | |
| MC | 1316801.88 | 1284078.38 | 1297748.25 | 2657888.00 | 2631427.25 | 2633413.50 |
| QMC (lat.) | **12922.79** | **12922.31** | **12948.75** | **12931.28** | **12928.22** | **12945.27** |
| RQMC (l.) | 1318297.38 | 1299869.75 | 1365709.75 | 2606723.50 | 2615697.50 | 2529304.00 |

(i) values for Laplace $(8 \times 8)$

|  | none | $f(x)$ | LOO | none | $f(x)$ | LOO |
|---|---|---|---|---|---|---|
|  | regular | | | antithetic | | |
| MC | 2641.38 | 8.15 | 8.15 | 16.28 | 16.27 | 16.29 |
| QMC (lat.) | **8.04** | **8.05** | **8.06** | 16.01 | 16.00 | 16.04 |
| RQMC (l.) | **8.09** | **8.09** | **8.10** | 16.19 | 16.17 | 16.22 |

(j) values for Laplace $(12 \times 12)$

|  | none | $f(x)$ | LOO | none | $f(x)$ | LOO |
|---|---|---|---|---|---|---|
|  | regular | | | antithetic | | |
| MC | 13593.82 | **41.40** | **41.45** | 82.73 | 82.71 | 82.92 |
| QMC (lat.) | **41.06** | **41.07** | **41.16** | 81.78 | 81.75 | 81.92 |
| RQMC (l.) | **41.32** | **41.31** | **41.36** | 82.62 | 82.64 | 82.80 |

(k) values for Triangular $(8 \times 8)$

|  | none | $f(x)$ | LOO | none | $f(x)$ | LOO |
|---|---|---|---|---|---|---|
|  | regular | | | antithetic | | |
| MC | 3090.80 | 10.21 | 10.11 | 20.27 | 20.43 | 20.07 |
| QMC (lat.) | **5.57** | **5.57** | **5.57** | 10.17 | 10.18 | 10.20 |
| RQMC (l.) | 884.22 | 9.88 | 9.82 | 19.14 | 19.71 | 19.76 |

(l) values for Triangular $(12 \times 12)$

|  | none | $f(x)$ | LOO | none | $f(x)$ | LOO |
|---|---|---|---|---|---|---|
|  | regular | | | antithetic | | |
| MC | 15975.60 | 49.73 | 49.89 | 99.81 | 99.32 | 100.31 |
| QMC (lat.) | **28.28** | **28.28** | **28.34** | 51.79 | 51.79 | 51.86 |
| RQMC (l.) | 4606.71 | 49.01 | 49.47 | 98.56 | 98.66 | 98.01 |

