# OpenReview forum: "Generalizing Stochastic Smoothing for Differentiation and Gradient Estimation"
_ICLR.cc/2025/Conference — Submitted to ICLR 2025_

### Official Review · Reviewer_RmT6 · 2024-10-24

**Soundness:** 3
**Presentation:** 4
**Contribution:** 3
**Rating:** 6
**Confidence:** 3

**Summary:**

The work addresses gradient estimation for stochastic differentiable relaxations of non-differentiable functions. The developed theory does not require a differentiable density or full support as it was in prior works.

**Strengths:**

The work contains rigorous theorem statements and proofs. The work contains extensive empirical results.

**Weaknesses:**

The work considers the problem of estimating the gradient of the relaxation of the algorithm or of the loss function. In Section 2.3 the authors formalize the algorithm smoothing and the objective smoothing. It seems that the paper does not provide the theoretical convergence analysis of the backbone stochastic optimization algorithms, but only introduces the techniques of smoothing and variance reduction and then tests the theoretical findings in practice.

**Questions:**

Is it possible to demonstrate the advantage of your smoothing approach over the previous works in theory, not only in practice, in some way? For instance, by incorporating the proposed gradient estimator into the SGD or Adam optimizer and providing a better convergence result?

There exist many general variance reduction techniques in the field of optimization (in such algorithms as SVRG, L-SVRG, Page, SAGA, Variance Reduced Adam, SPIDER etc.). Do the proposed variance reduction techniques generalize the techniques from the optimization field or orthogonal to them? Can they be combined together?

Smoothing should introduce some bias to the estimator. There are some works on the analysis of SGD with biased estimators [1, 2]. They both consider smoothing techniques. Does the proposed smoothing technique fit into the frameworks of assumptions in these papers?

[1] Ahmad Ajalloeian, Sebastian U. Stich, On the Convergence of SGD with Biased Gradients
[2] Yury Demidovich, Grigory Malinovsky, Igor Sokolov, Peter Richtárik, A Guide Through the Zoo of Biased SGD

---

> ### Author Response · Authors · 2024-11-20
> **Response to Reviewer RmT6 [1/2]**
>
> Dear Reviewer RmT6,
>
> Thank you very much for reviewing our work and particularly for your intruiging remarks that lead us to compare the constants for the variances, and thus convergence rates in optimization.
> Also, thank you for appreciating our "rigorous theorem statements and proofs" as well as "extensive empirical results".
>
> > "theoretical convergence analysis of the backbone stochastic optimization algorithms"
>
> Yes, indeed we do not provide theoretical convergence analyses of the backbone stochastic optimization algorithms.
> This is because our focus lies on the estimation of the gradients, with the goal of then backpropagating those gradients into an upstream neural network.
> But in this context, we quantify the variance of the gradient estimates, in dependence on the number of samples, which can be simply plugged into the convergence rates of respective individual choices of backbone stochastic optimization algorithms.
>
> **Questions:**
>
> > "Is it possible to demonstrate the advantage of your smoothing approach over the previous works in theory, not only in practice, in some way?"
>
> Between the choice of distribution, and without a variance reduction technique, the asymptotic convergence / variance reduction rate (under assumption of finite variance in relation between $f$ and $\mu$) is always the same: $\mathcal{O}(1/s)$.
> However, as we discuss below, the constants differ between different distributions.
>
> With RQMC, which has not previously been considered in the area of stochastic smoothing, we improve the variance reduction rate to $\mathcal{O}(1/s^{1+2/n})$.
>
>
> Variances in the scenario of the sign function ($f(x)=\operatorname{sign}(x)$), smoothed with each respective distribution, and evaluating the variance at point $0$, in dependence of the number of samples $s$.
> To standardize the distributions, we consider 2 settings: scaling each input distribution to a variance of 1, as well as choosing a scale such that the Lipschitz constant of the resulting smooth function is 1.
> We want to point out that these results directly line up with our analytical computation of the variances.
>
> | $f(x)=\operatorname{sign}(x)$ | Var=1 | Lipschitz=1 |
> |--------------|-----------|-------------|
> | Gaussian     | $0.364/s$ | $0.573/s$   |
> | Logistic     | $0.272/s$ | $0.336/s$   |
> | Gumbel       | $0.759/s$ | $0.859/s$   |
> | Cauchy       | n/a       | $0.234/s$   |
> | Laplace      | $0.000/s$ | $0.000/s$   |
> | Triangular   | $1.747/s$ | $2.496/s$   |
>
> Here, the Laplace distribution performs best, consistenty achieving a variance of 0, which however, in this particular case is due to the simple nature of the problem. The second best distribution is the logistic distribution. The logistic behavior is similar to the Laplace distribution also from an analytical perspective (the logistic can be seen as a smoothed variant of the Laplace).
>
> To introduce a more complex setting, we further consider the sign of the sine ($f(x)=\operatorname{sign}(\sin(x))$):
>
> | $f(x)=\operatorname{sign}(\sin(x))$ | Var=1 | Lipschitz=1 |
> |--------------|-----------|-------------|
> | Gaussian     | $0.382/s$ | $0.588/s$   |
> | Logistic     | $0.318/s$ | $0.690/s$   |
> | Gumbel       | $0.840/s$ | $1.130/s$   |
> | Cauchy       | n/a       | $0.452/s$   |
> | Laplace      | $0.093/s$ | $0.144/s$   |
> | Triangular   | $1.727/s$ | $2.609/s$   |
>
> Here, we observe a very similar behavior, but with the Laplace distribution having a variance different from $0$---still it has the smallest constant for the variance of the gradient estimator.
>
> Additionally, we consider the inital setting ($f(x)=\operatorname{sign}(x)$) also with RQMC (latin) at $s=100$ samples, which drastically reduces the variance further. As the rate is faster than $1/s$, we report it for 100 samples.
>
> | $f(x)=\operatorname{sign}(x)$ | RQMC @s=100, Var=1 | RQMC @s=100, L=1   |
> |--------------|--------------------|--------------------|
> | Gaussian     | $0.0000224$        | $0.0000355$        |
> | Logistic     | $0.0000011$        | $0.0000013$        |
> | Gumbel       | $0.0001780$        | $0.0002008$        |
> | Cauchy       | n/a                | $0.0000041$        |
> | Laplace      | $0.0000000$        | $0.0000000$        |
> | Triangular   | $0.0109100$        | $0.0156400$        |

---

> ### Author Response · Authors · 2024-11-20
> **Response to Reviewer RmT6 [2/2]**
>
> > "For instance, by incorporating the proposed gradient estimator into the SGD or Adam optimizer and providing a better convergence result?"
>
> It is not clear to us to which prior convergence result you would like us to compare our zeroth order gradient estimation methods. While plugging in our variances is simple, having a variance in a zeroth-order estimator will always be larger than an exact analytical first-order gradient computation with variance 0. Further, when comparing to the first-order gradients of functions that have no defined non-zero gradients, our method is clearly superior as SGD and Adam both fail if the gradient is almost always 0 (and NaN/infinite otherwise).
>
> Finally, we would like to point out that the source of variance in SGD for neural network training is typically a different one from the one we have: in SGD, the variance, by default, arises from the random choice of batch. In stochastic smoothing, we have a variance within the gradient estimation step.
>
> > "SVRG, L-SVRG, Page, SAGA, Variance Reduced Adam, SPIDER"
>
> The listed VR techniques from the field of optimization are orthogonal to our VR techniques.
> Their VR techniques use stochastic gradients and attempt to find the best update step in consideration of prior gradients (e.g., for other batches of data)---in contrast, our VR techniques are for the gradient estimation itself, and are applicable before the gradient has been computed.
> Our gradient estimates can be used to optimize an upstream network with algorithms as SVRG, L-SVRG, Page, SAGA, Variance Reduced Adam, SPIDER etc., so, yes, they can be combined. But this is of course only favorable if $f$ is not differentiable and its gradient not analytically computable.
>
> > "bias to the estimator"
>
> While, from the perspective you are coming from, the smoothing introduces some bias to the estimator, it is important to keep in mind that the gradient of the original function is often zero almost everywhere and undefined elsewhere. In that context the function whose gradient we want to estimate is $f_\epsilon$, and for this our gradient estimators are unbiased. (In this context, AlgoVision [5] can be viewed as / is motivated as an attempt at evaluating $f_\epsilon$, but via a closed-form approximation, which has a zero-variance, but introduces a bias due to approximation.)
>
> If one viewed it from the perspective of the bias between the true gradient of $f$ and the gradient of $f_\epsilon$, then that bias is unbounded / undefined by definition as soon as $f$ is discontinuous, because $f_\epsilon$ has a finite defined gradient and $f$ has an undefined gradient at discontinuities.
> Such bias conceptually does not fit in the frameworks of assumptions of [1, 2].

---

> ### Comment · Reviewer_RmT6 · 2024-11-26
>
> I thank the Authors for the detailed response!
> I will keep my score.

---

### Official Review · Reviewer_FXvh · 2024-11-02

**Soundness:** 2
**Presentation:** 2
**Contribution:** 1
**Rating:** 3
**Confidence:** 4

**Summary:**

The authors study stochastic gradients, with applications in stochastic smoothing and variance reduction of gradient estimation.

I recommend rejection of this paper because I feel there is a lack of novelty and impact for machine learning.

I present a more detailed summary of the paper along with my criticisms under the weakness section.

**Strengths:**

The experiments in Section 4 seem quite interesting, particularly the use of smoothing for relaxing discrete functions to continuous functions.

**Weaknesses:**

The goals of the paper are somewhat diverse, so I will go over each result and state my criticisms.

1. In Lemma 3, the authors present a result for smoothing an objective $f$ using a distribution $\mu$ that is possibly non-differentiable. However, the non-differentiable set for $\mu$ is assumed to be $0$-measure. In any learning application I can think of, there seems to be no meaningful difference between "differentiable" and "almost everywhere differentiable". For instance, one can always convolve $\mu$ with a tiny Gaussian, of variance less than machine precision, and immediately get a differentiable density, while not affecting the outcome in any meaningful way. There are also no quantitative analysis which indicate that choosing a non-differentiable $\mu$ would offer any advantages.
2. Theorem 7 discusses smoothing with an additional rescaling matrix involved. The result appears unsurprising, and the proof appears to use just calculus. I think the authors want to apply this to finding a optimal rescaling matrix $\mathbf{L}$, but the consequences of Theorem 7 are never discussed.
3. The authors discuss numerous variance reduction techniques in Section 2.1, but there is no attempt at any quantitative and theoretical comparison of these approaches. The authors did include some simple synthetic experiments, but it is questionable how well the observations will generalize to realistic scenarios.
4. In Section 2.3, the authors discuss smoothing of the algorithm vs the objective. I am not convinced by the setup of $\ell(h(y))$ where $\ell$ is the loss, $h$ is an algorithm, and $y$ is the model output.
5. In Section 4, the authors present a series of experiments that compare the effectiveness of different distributions and different variance reduction techniques. I am not sure how well the conclusions here generalize.

**Questions:**

1. Can the authors comment on how their theoretical or experimental results might be applicable for discrete-to-continuous relaxations? Particularly, how would quantities such as lipschitz smoothness, convexity, or general ease of optimization be affected by different choices of distributions?

2. Can the authors comment on the run-time of their algorithms compared to the baselines in Section 4?

---

> ### Author Response · Authors · 2024-11-20
> **Response to Reviewer FXvh [1/2]**
>
> Dear Reviewer FXvh,
>
> Thank you very much for reviewing our work and for finding our experiments interesting.
> We especially appreciate your thought-provoking thought experiment of mollifying the density of the used distribution to make it formally differentiable.
>
> > 1. "In any learning application I can think of, there seems to be no meaningful difference between "differentiable" and "almost everywhere differentiable"."
>
> We would like to point out that there are in fact major differences between “differentiable” and “almost everywhere differentiable” in learning applications. For example, the Heaviside function is differentiable almost everywhere, but not differentiable. Indeed, the gradient of the Heaviside is *zero* almost everywhere and undefined elsewhere, making this gradient completely useless in training. Almost everywhere differentiable is an extremely weak condition, and it does not imply any existence of non-zero gradients.
>
> > 1. "For instance, one can always convolve $\mu$ with a tiny Gaussian, of variance less than machine precision"
>
> While, technically, one can always convolve or mullify the density $\mu$ used for smoothing with a tiny Gaussian, if the characterization given by Lemma 3 (the continuity) is not met, it may not lead to helpful gradients. Taking the example of a “tiny Gaussian, of variance less than machine precision” and let us use the uniform ($[0, 1]$) density. Now, in the range $(0, 1)$, i.e., in the vast majority of the probability mass, or with high probability at all points of a finite set, the derivative of the NLL ($\nabla_\epsilon \log \mu (\epsilon)$) is zero to numerical precision. Thus, the empirical gradient estimate will be zero with high probability for finite numbers of samples. If however, samples outside of $(0, 1)$ are drawn, these have derivatives of the NLL that are explodingly large (inverse of numerical precision), and will in practice be NaN or inf.
>
> One may argue that the variance does not need to be low towards numerical precision, but if the variance is moderately small, then the gradient estimator will still have an extremely high variance (most samples having a derivative of NLL close to zero, and few having a derivative of NLL proportional to the inverse of the Gaussian’s standard deviation.) This leads to a high variance estimator, which is problematic in practice.
>
> So, without the (absolute) continuity, which our Lemma 3 shows to be a sufficient condition, even with smoothing, it is not practically possible to perform stochastic smoothing with arbitrary non-differentiable densities, even when mullified.
>
> We find this thought experiment interesting and offer to include an extended remark in the appendix of the camera-ready, or in the main paper in case we are allowed an additional page.
>
> > 2. "Theorem 7 ..."
>
> We would like to remark that this result is to our knowledge not previously know and thus novel; if you have a reference, please let us know. “use just calculus” appears to be an odd criticism. Also, we should remark that this Theorem builds upon the other Lemmas, and the proofs of the used Lemmas are of course not repeated. Yes, indeed, we apply this to finding an optimal rescaling matrix; space was limited, and we wanted to avoid submitting an even longer paper, so we included a sentence of discussion as a compromise.
>
> > 3. "The authors discuss numerous variance reduction techniques in Section 2.1..."
>
> We provide theoretical comparison of the approaches as far as applicable around line 250. For the constants in different settings, please see our response to Reviewer RmT6.
>
> Throughout our experiments, we provide various quantitative evaluations, and these are on **the** standard benchmarks of the respective communities: Differentiable sorting and ranking operations are always (refs [7, 8, 9, 11, 12], published at NeurIPS 2019, ICLR 2019+2022, ICML 2020+2021) benchmarked primarily on the 4-digit MNIST sorting benchmark (*not* to be confused with MNIST, this is a very different data set, and very different supervision in the form of ordering, and no absolute supervision.)
> For differentiable shortest-path algorithms, **the** standard benchmark is the Warcraft shortest-path benchmark (refs [2, 5, 17], published at ICLR 2020, NeurIPS 2020+2021).
>
> > 4. "In Section 2.3..."
>
> The setup of $\ell(h(y))$ is a standard setup in algorithmic supervision, and a setup shared between almost all of our references regarding differentiable relaxations, in particular diff. sorting, diff. top-k, diff. shortest-path, diff. simulators, diff. rendering, etc. Since the setup is well-established, we are not sure what you mean by saying that you are not convinced by it.

---

> ### Author Response · Authors · 2024-11-20
> **Response to Reviewer FXvh [2/2]**
>
> > 5. "In Section 4, the authors present a series of experiments that compare the effectiveness of different distributions and different variance reduction techniques. I am not sure how well the conclusions here generalize."
>
> Thanks. Given that our variance reduction results are consistent between all the different experiments, we strongly expect that they would generalize even onto other domains beyond the 4 examined. As for the choice of distribution, this is domain specific, for example for differentiable rendering, it has been shown that the choice of distribution even depends on the 3d shapes of the objects [22], and for differentiable sorting it has been shown that the Cauchy distribution is advantageous in differentiable sorting networks [12]. We observed the expected effects, and enabled new optimal settings, illustrating that the conclusions generalize well.
>
> **Questions:**
>
> > 1. "Can the authors comment on how their theoretical or experimental results might be applicable for discrete-to-continuous relaxations? Particularly, how would quantities such as lipschitz smoothness, convexity, or general ease of optimization be affected by different choices of distributions?"
>
> In accordance with the nomenclature of the community of differentiable relaxations and simulators, our experimental results do indeed lie in the space of discrete-to-continuous relaxations.
> (In most cases our functions are $f: R^n\to \\{0, 1\\}^m$.)
>
> If you are referring to components like Gumbel-Softmax, we have since submission also used our method in this direction, enabling, e.g., attention mechanisms with other distributions for transformers as well as discrete VAEs with promising results, but this lies outside the scope of the current work considering that our paper is already very dense and has many results deferred to the appendix.
>
> Regarding Lipschitzness, for functions $f: R^n\to \\{0, 1\\}^m$, we provide Lipschitz constants in the following for each distribution:
>
>
> | Dist.        | Lipschitz constant |
> |--------------|-----------|
> | Gaussian     | $\frac{1}{\sqrt{2\pi}\gamma} \approx 0.399/\gamma$  |
> | Logistic     | $0.25/\gamma$  |
> | Gumbel       | $\frac{1}{e\gamma} \approx 0.368/\gamma$  |
> | Cauchy       | $\frac{1}{\pi\gamma} \approx 0.318/\gamma$  |
> | Laplace      | $0.5/\gamma$  |
> | Triangular   | $1.0/\gamma$  |
>
> Regarding convexity, convexity is usually in conflict with the desired behavior of $f$. In edge cases, convexity of $f_\epsilon$ from non-convex $f$ may occur, but this is not the intend of our work, does not occur in the examined scenarios, and the desired behavior of $f_\epsilon$ actually typically prohibits convexity.
>
>
> > 2. "Can the authors comment on the run-time of their algorithms compared to the baselines in Section 4?"
>
> For sorting/ranking, the baseline runtimes range from 67 to 78 minutes. Accordingly, 1024-8192 samples with our method can be afforded in the computational budget of the baseline methods.
> This comparison excludes entropically regularized optimal-transport-based sorting (OT.S.) for which our implementation is inefficient, taking tens of hours; faster implementations of the underlying routines would be available in JAX but are not compatible with our PyTorch-based codebases.
>
> For shortest-path, the AlgoVision baseline requires 15 minutes, which corresponds to roughly 50 samples.
> The stochastic smoothing-based Fenchel-Young (FY) loss with 1 sample required 159 minutes, where the cost is dominated by the cost of running the Dijkstra algorithm.
> By utilizing a new highly parallelized Bellman-Ford implementation, our method with 10 000 samples is still faster than 1 sample with prior implementations.
> For fairness, we want to point out that our efficient & highly parallelized Bellman-Ford implementation could equally be used for other Blackbox approaches making them per sample as efficient in runtime as ours. --- But this illustrates an important point of our paper: it can be very fruitful to accelerate $f$ in order to afford a large number of samples, a regime where our method really shines, and which was not explored in prior works.
>
> In prior works, the largest employed number of samples considered for any shortest-path Blackbox-based approaches was 30, and no Blackbox-based approach was able to compete with AlgoVision, something that we changed by leveraging 10 000 samples in a runtime shorter than previously a single sample.
>
> For the rendering example, a meaningful runtime comparison is not available because, for scientific accuracy and consistency, we use a differentiable renderer in "hard mode", which is a very compute-inefficient approach (which was irrelevant for the purpose of scientific study) but enables a perfectly consistent comparison.
> For the differentiable cryo-ET example, the runtime is exclusively dominated by the simulator time, making any runtime comparison directly proportional to the number of samples (x axis in Figure 9).

---

> > ### Comment · Reviewer_FXvh · 2024-11-21
> >
> > > We would like to point out that there are in fact major differences between “differentiable” and “almost everywhere differentiable” ...
> >
> > Yes, in mathematical analysis there is a difference. My point is that in machine learning there is no meaningful difference. See below.
> >
> > > If however, samples outside of are drawn, these have derivatives of the NLL that are explodingly large (inverse of numerical precision), and will in practice be NaN or inf.
> >
> > I think you misunderstand my original point. Allow me to rephrase:
> >
> > Let $\epsilon$ denote machine precision. Let $\delta$ denote a value far below machine precision. Your algorithm will only ever evaluate gradients at values of $k\epsilon$, for integers $k$. Therefore, in the machine implementation of your algorithm, **it makes no difference whether you use $\mu$ or the $\delta$-mollified $\mu$, every gradient step is identical.** You do not worry about nan or inf at some points $x$, **because you never see those points**, just as you never see points in $\Omega$.
> >
> > Now there is a catch to my reasoning above, partially related to your following comment:
> >
> > >  variance does not need to be low towards numerical precision, but if the variance is moderately small, then the gradient estimator will still have an extremely high variance
> >
> > Whereas the estimator for $\nabla \mu_{\delta}$ may incur a large point-wise (and possibly $L_2$) error, the estimator for $\nabla_x f_\epsilon(x)$ (which is the target you care about) *does not*. There are numerous ways to show this. The simplest way to convince yourself is again by appealing to the above claim -- that the algorithm with $\mu_\delta$ proceeds exactly as it would with $\mu$. But even with a mathematical approach, without appealing to numerical precision, one can cap the gradient at a suitable constant, instead of having it scale to a unreasonably large value of $1/\delta$ (for reasonable $f$).
> >
> > I thus claim the following: **for any algorithmic error bound you derive for lemma 3 using a.c. $\mu$, the same error bound can be derived with mollified $\mu_\delta$**. At the moment, you have no concrete error bounds for Lemma 3 under finite computation, nor error bounds for any downstream algorithm (for this you will likely need additional assumptions on $f$); thus it is difficult for me to state an alternative proof with $\mu_\delta$, but I welcome further discussion on this issue.
> >
> > > We would like to remark that this result is to our knowledge not previously know and thus novel; ... “use just calculus” appears to be an odd criticism.
> >
> > "use just calculus" is indeed a vague criticism, and I apologize for the vagueness. Let me state my more precise crriticism below:
> >
> > The proof of Theorem 7 goes from equation (36) to (61).
> > 1. (36)-(37) is simply change-of-variable formula for volume.
> > 2. (38)-(39) is interchanging derivative with integral.
> > 3. (40) is matrix chain rule
> > 4. (41) is change-of-variable for volume
> > 5. (42)-(43) is basic algebra manipulation.
> > 6. (44)-(49) again uses the same tools as above, with different order.
> > 7. (50)-(52) is literally just matrix chain rule.
> > 8. (53) is plugging the formula for gradient of determinant
> > 9. (54)-(61) is basic algebra
> >
> > Given the above, my issue is that **the proof of Theorem 7 is extremely simple**, the tools used are just basic chain rule, matrix calculus, and change-of-variables. The **steps taken are very straightforward**, so I think most people will have no problem writing the proof from scratch.
> >
> > > if you have a reference, please let us know
> >
> > I disagree that the result should be considered novel just because it is not in the literature. It is also entirely possible that no one thought necessary to write it down as a theorem.
> >
> > > We provide theoretical comparison of the approaches as far as applicable around line 250
> >
> > The theoretical comparison on line 250 seems to be simply quoting rates from prior work. My question on variance reduction is **what is your contribution here**? From my understanding, it seems that you
> > 1. took some existing VR methods,
> > 2. plugged them in to the estimator,
> > 3. cited existing variance bounds
> > 4. evaluated them experimentally
> >
> > Am I missing any constribution related to variance reduction?
> >
> > > The setup of $\ell(h(y))$ is a standard setup in algorithmic supervision
> >
> > I see. I suggest you state that this is motivated by "algorithmic supervision" instead of "in many learning problems" as it currently states on line 269. I do not mean to nitpick but it is quite confusing for me as I assumed you are talking about general machine learning problems.

---

> > > ### Comment · Reviewer_FXvh · 2024-11-21
> > >
> > > > Given that our variance reduction results are consistent between all the different experiments, we strongly expect that they would generalize even onto other domains beyond the 4 examined
> > >
> > > Sounds reasonable.
> > >
> > > > on discrete-to-continus relaxations.
> > >
> > > Thank you for the explanation.
> > >
> > > > on runtime
> > >
> > > Thank you for the details. It is understandable that some baselines are faster due to using better optimized packages.
> > >
> > > Were there any case where you got a speed-up (or some other kind of measurable advantage, e.g. being able to sample more efficiently) **as a result of your theory**?

---

> ### Author Response · Authors · 2024-11-22
>
> Thank you for your response!
>
> We are a bit unsure whether your argument wrt. a mollified $\mu$ is for differentiable almost everywhere $\mu$ or for absolutely continuous $\mu$.
>
> Assuming your argument should apply to differentiable almost everywhere $\mu$, then it should apply to the Uniform ([0, 1]) density.
> For this, let us first lay out each of the derivatives of NLLs in this case:
> $$
> \nabla_\epsilon - \log \mu (\epsilon) = \begin{cases}
>     \text{undefined} (-\infty) & \epsilon=0 \\\\
>     0 & 0<\epsilon<1 \\\\
>     \text{undefined} (+\infty) & \epsilon=1
> \end{cases}
> $$
> Even when mollifying and clamping, between 0 and 1, $\nabla_\epsilon - \log \mu (\epsilon)$ will numerically still be 0.
> But at 0 and 1, it is undefined / infinite due to the discontinuity of the density in $\mathbb{R}$, and even when clamping it will just be a large value that never shows up in the empirical estimator.
> Thus, with any number of samples, the finite sample estimator (7) will result in zero almost always, which is an incorrect estimate of the gradient.
>
> You may have observed that, e.g., smoothing a Heaviside with a uniform distribution results in a function that is constant zero, then linear (slope 1), then constant 1. This function is differentiable almost everywhere, but *importantly* even when doing everything to stabilize the gradient estimator, the stochastic smoothing gradient estimator will be utterly useless for any attempt at estimating this gradient because it will always return a gradient of zero.
>
> We understand that this may not become obvious from Remark 4, for which we apologize, and offer to add an additional discussion to the paper.
>
> If your entire argument should only apply to absolutely continuous $\mu$, then we believe it to be more of a suggestion for a more complex alternative proof (assuming this proof route works and doesn't fail on a technicality). This would mean that the characterization for requirement of absolute continuity, first proposed in our Lemma 3, is valuable as it precisely characterizes the same requirements as your proposal would require.
>
> We are looking forward to your response to better understand your perspective, and clear up the misunderstanding.
>
> ---
>
> > "At the moment, you have no concrete error bounds for Lemma 3 under finite computation", "will likely need additional assumptions on $f$"
>
> If you would like us to provide concrete error bounds for Lemma 3 for the different choices of distributions under finite computation for restricted $f$, we are happy to include them (we didn't include them as we didn't consider them sufficiently relevant to warrant the additional space they take up.)
>
> > "constribution related to variance reduction"
>
> Your list of contributions fits, but we want to emphasize that no prior work used (R)QMC for stochastic smoothing, and that our evaluation is particularly extensive and insightful, finding that (R)QMC is consistently best and far superior over the popular antithetic sampling.
>
> > "I see. I suggest you state that this is motivated by "algorithmic supervision" instead of "in many learning problems" as it currently states on line 269. I do not mean to nitpick but it is quite confusing for me as I assumed you are talking about general machine learning problems."
>
> Thank you for the suggestion, we agree and have adjusted it as per your suggestion in the revision.
>
> > "It is understandable that some baselines are faster due to using better optimized packages."
>
> If you were referring to our OT.S. discussion, we wanted to add that OT.S. is still more computationally expensive than the other baselines (already from first principles because it requires many iterations for solving OT with Sinkhorn), we just cannot give a fair exact time comparison because the fastest available implementation is not compatible. When saying "our implementation [of OT.S.] is inefficient", it was actually that the reference implementation of OT.S. is slow, and there are faster implementations of entropically regularized optimal transport available.
>
> > "Were there any case where you got a speed-up (or some other kind of measurable advantage, e.g. being able to sample more efficiently) as a result of your theory?"
>
> On differentiable sorting, for smaller numbers of samples (256 & 1024), the best performance is achieved using the Laplace distribution, a distribution that is supported from our theory, and, by weighting all samples equally, has the lowest variance in the gradient estimator.
>
> While not strictly theory, by proposing the concept of *smoothing the algorithm*, we achieve substantial improvements on the Warcraft shortest path benchmark, requiring over 10x fewer samples to reach comparable accuracy (Fig. 14, smoothing the *algo*rithm with 1000 samples is better than smoothing the *loss* with 10000 samples). (*Smoothing the loss* is the default case.)

---

> ### Comment · Reviewer_FXvh · 2024-11-22
>
> > ... smoothing a Heaviside with a uniform distribution results in a function that is constant zero, then linear (slope 1), then constant 1. This function is differentiable almost everywhere, but importantly even when doing everything to stabilize the gradient estimator, the stochastic smoothing gradient estimator will be utterly useless for any attempt at estimating this gradient because it will always return a gradient of zero.
>
> I do not think your Lemma 3 applies to the Heaviside function, because it does not obey the fundamental theorem of calculus over $\Omega^c$? **Why do you keep using the Heaviside function as an example if Lemma 3 does not even apply to it?**
>
> **Edit shortly after my original post**: I noticed that in my original comment I mentioned “differentiable” and “almost everywhere differentiable”. Perhaps this was why you brought up the heaviside function? **I should have said "differentiable" and "absolutely continuous"**; I apologize for the imprecise choice of words (it was because I assumed that it would be obvious in the context of our discussion of Lemma 3). *If this is the reason for bringing up the Heaviside function*, then please ignore my comment above. **My original point is that there is no meaning in distinguishing a.c vs differentiable for machine learning purposes**. I think the discussion immediately below concerns this:
>
> > If your entire argument should only apply to absolutely continuous $\mu$, then we believe it to be more of a suggestion for a more complex alternative proof (assuming this proof route works and doesn't fail on a technicality). This would mean that the characterization for requirement of absolute continuity, first proposed in our Lemma 3, is valuable as it precisely characterizes the same requirements as your proposal would require.
>
> My claim, based on the "proof" I sketched, is that **in any machine learning application, the replacement of a.c. $\mu$ by a mollified (and hence differentiable) $\mu_\epsilon$ works for (9) *trivially*** I disagree that what I said is a "more complex alternative proof". In fact, I think it is a stretch to even call what I presented "a proof" because **it is so trivial**. And it is because the argument is so trivially simple that I do not see the purpose of distinguishing between a.c. and differentiable (again, for machine learning, not for mathematical analysis).
>
> Also there is no need to "assuming this proof route works and doesn't fail on a technicality". As I said in the previous comment, **the algorithm will proceed identically** on $\mu$ and $\mu_\epsilon$ when $\epsilon$<<machine precision.
>
> > If you would like us to provide concrete error bounds for Lemma 3 for the different choices of distributions under finite computation for restricted $f$, we are happy to include them (we didn't include them as we didn't consider them sufficiently relevant to warrant the additional space they take up.)
>
> I am not asking for a concrete error bound, feel free to leave things as it is. What I meant in my original statement is: if you would like to have a more quantitative debate on "whether mollification suffices", then we will need to discuss concrete bounds (under specific constraints).
>
> > On differentiable sorting, for smaller numbers of samples (256 & 1024), the best performance is achieved using the Laplace distribution, a distribution that is supported from our theory, and, by weighting all samples equally, has the lowest variance in the gradient estimator.
>
> That makes sense.
>
> However, this is again tied to my first issue: If you simply smoothed the function at $0$ with a very tiny gaussian, I suspect that your performance would not have changed at all (in fact if the gaussian is small enough, your algorithm output should be identical). You *will* incur an extremely large hessian at $0$, but the consequences of that are orthogonal to the present discussion.

---

> ### Author Response · Authors · 2024-11-22
>
> Thank you for your response!
>
> > I do not think your Lemma 3 applies to the Heaviside function, because it does not obey the fundamental theorem of calculus over $\Omega^c$? Why do you keep using the Heaviside function as an example if Lemma 3 does not even apply to it?
>
> Lemma 3 applies to it, we use the Heaviside function as an example for $f$. The Heaviside function lies in the space of allowed functions $f:\mathbb{R}^n\to\mathbb{R}^{(m)}$ and provides similarities to the majority of functions we also consider in the experiments, most of which are in $f: \mathbb{R}^n\to \\{0, 1\\}^m$.
>
> Smoothing a continuous, absolutely continuous, or already differentiable function is a different and easier problem, solvable with first-order gradient estimators; our functions $f$ are specifically discontinuous.
>
> To prevent misunderstandings, **could you confirm whether you agree that the Heaviside function smoothed with a Uniform distribution is not possible with the stochastic smoothing gradient estimator (even with a tiny Gaussian mollified density)** because the Uniform distribution has a discontinuous density? We are happy to elaborate on this.
>
> ---
>
> With "suggestion for a more complex alternative proof" we mean that providing a formal proof of differentiability of $f_\epsilon$ as well as exactness of the gradient estimator with a mollified a.c. $\mu$ requires more than simply taking the limit of $\sigma\to0$.
> For example, one consideration that would be lost is ensuring that the derivative of the negative log likelihood in regions of $\mu=0$ and $\mu_\epsilon>0$ actually disappears and does not explode.
> Also, if density $\mu$ has bounded range, and is mollified with a smooth density, the resulting $\mu_\epsilon$ is differentiable, but that must not mean that it may be plugged into Lemma 1, because that would imply that the Uniform distribution would be allowed for $\mu$, constructing a counter-example for such a proof.

---

> > ### Comment · Reviewer_FXvh · 2024-11-22
> >
> > > Lemma 3 applies to it, we use the Heaviside function as an example for $f$. The Heaviside function lies in the space of allowed functions.
> >
> > Now I am really confused. We have always been discussing the whether your claim of handling **a.c. $\mu$** is a significant improvement over **"differentiable $\mu$"**. The differentiability (or absolute continuity, or any other property) of $f$ **was NEVER a point of contention**. Am I missing something in our discussion?
> >
> > **Again, my point is that going from "$\mu$ is differentiable" to "$\mu$ is a.c." is not a meaningful improvement.**
> >
> > > To prevent misunderstandings, could you confirm whether you agree that the Heaviside function smoothed with a Uniform distribution is not possible with the stochastic smoothing gradient estimator (even with a tiny Gaussian mollified density) because the Uniform distribution has a discontinuous density? We are happy to elaborate on this.
> >
> > I can confirm the following: for the random smoothing expression (9) in your paper, if $\mu$ is the uniform distribution, (9) will not work as the rhs is always $0$.
> >
> > >  because that would imply that the Uniform distribution would be allowed for $\mu$, constructing a counter-example for such a proof.
> >
> > Again, I am not sure why you bring up $\mu$=Uniform, which is **not a.c.** as an (counter)-example? Because it is not a.c, **it is not handled by Lemma 3**.

---

> ### Author Response · Authors · 2024-11-24
>
> Let us try to recapitulate the state of our discussion. Our claim is that, in order to be useable for stochastic smoothing, a density $\mu$ must be absolutely continuous. This is meant as a characterization of the admissible densities. Your claim is that this is no meaningful improvement over simply differentiable (everywhere), because any absolutely continuous density can be mollified to make it differentiable (by convolving with a very narrow Gaussian), and in such a way that in the discrete space of numbers representable in a computer no numeric difference results. Our counterargument is that such a mollification can just as well be applied to discontinuous densities like the uniform density, making it differentiable as well. However, as far as we gather from our discussion so far, you agree that the uniform density is not a feasible density for stochastic smoothing. Hence we are wondering what your characterization of the densities feasible for stochastic smoothing is. Due the above, it cannot be that mollification makes them differentiable ("mollified (and hence differentiable)"), because that would also apply to the uniform density. If, on the other hand, absolute continuity is actually required, then our Lemma 3 provides the relevant characterization of feasibility for stochastic smoothing.

---

> > ### Comment · Reviewer_FXvh · 2024-11-24
> >
> > > Our counterargument is that such a mollification can just as well be applied to discontinuous densities like the uniform density, making it differentiable as well. However, as far as we gather from our discussion so far, you agree that the uniform density is not a feasible density for stochastic smoothing.
> >
> > That is a **invalid counter argument**. We are trying to distinguish between **different** vs **absolutely continuous**. My point is that your Lemma 3 has **no meaningful contribution** because any absolutely continuous **\mu** can be treated as if mollified, at no additional cost.
> >
> > You raise as counter example uniform density $\mu$ which is **DISCONTINUOUS**. It is obvious that the uniform density would not work with (9) because **the way you define stochastic smoothing** makes no sense for a discontinuous density.
> >
> > A valid counter argument would be stating a density **\mu**, that satisfies the following:
> > 1. is **absolutely continuous** but not differentiable.
> > 2. $\mu$ works with (9) in Lemma 3
> > 3. the mollified version $\mu$ does not work for (9).
> >
> > I claim that such a counter example does not exist. I do not have time to keep running in circles on the same question. This is the third or fourth time that I am repeating myself, you know exactly what I mean but insist on the completely irrelevant uniform distribution.
> >
> > I will be maintaining my score.

---

> ### Author Response · Authors · 2024-11-24
>
> We are **not** arguing that an a.c. $\mu$, when mollified, is prohibitive of working for stochastic smoothing.
> Lemma 3 already shows that a.c. $\mu$ works without necessitating mollification, and mollification should not break this.
>
> What we are showing is that mollifying a density, and thus making it differentiable cannot be sufficient to prove that it works for stochastic smoothing:
> The argument of mollifying a density and making it differentiable still applies, whether $\mu$ is a.c. or discontinuous.
> **(A differentiably mollified discontinuous density is differentiable.)**
> **Because the mollification argument would apply regardless of whether the density is a.c. or discontinuous, it cannot be a sufficient argument for a proof.**
> (If the condition (a.c.) is given but not needed in the proof, it must not be necessary. If a.c. is given, it can be mollified, the limit can be taken, and it can be applied to Lemma 3. And if we do not take the limit, we would also equally be able to mollify the uniform distribution.)
>
> While one can of course make an intuitive argument for mollification, we do not believe that it could lead to a simpler formal statement or proof route than Lemma 3.
>
> > "My point is that your Lemma 3 has no meaningful contribution because any absolutely continuous \mu can be treated as if mollified, at no additional cost."
>
> Other than it being stated in our Lemma 3, why would "absolutely continuous" be necessary in the mollification argument?
>
> (To illustrate the intricacies, we want to point out that even a simple "continuous" is not sufficient for Lemma 3 or stochastic smoothing in general, something that in our opinion does not become clear from the perspective of mollification without Lemma 3.)

---

> > ### Comment · Reviewer_FXvh · 2024-12-01
> >
> > > Because the mollification argument would apply regardless of whether the density is a.c. or discontinuous ... we would also equally be able to mollify the uniform distribution.)
> >
> > This reasoning above seems to be based on your claim from an earlier comment that
> > > Heaviside function smoothed with a Uniform distribution is not possible with the stochastic smoothing gradient estimator (even with a tiny Gaussian mollified density).
> >
> > This is **completely fallacious** because you are conflating **two notions of "not possible"**:
> > 1. Mathematical validity: the mollified uniform density $\mu_\epsilon$ **is differentiable**, and so (9) is mathematically defined. In contrast, $\mu$ is **mathematically invalid** for Lemma 3.
> > 2. Computational cost: estimation of $\nabla \mu_\epsilon$ may have high variance leading to high cost. However, **Lemma 3** **does not even guarantee a rate**.
> >
> > You argue that $\nabla \mu_\epsilon$ is a bad idea, when Lemma 3 cannot even handle $\mu$ (due to discontinuity, not lack of differentiability). In addition, Lemma 3 **gives no quantitative guarantee**, so even if $\mu_\epsilon$ involves some ridiculous constant like $10^{52}$, **it is still a better guarantee than Lemma 3.**
> >
> > So if $\mu$ were discontinuous, **mollification gives a stronger guarantee than Lemma 3**. If $\mu$ satisfies stronger conditions, **mollification still gives stronger guarantees than Lemma 3.** Thus the uniform $\mu$ does not demonstrate Lemma 3 is favorable over mollication.
> >
> > > Other than it being stated in our Lemma 3, why would "absolutely continuous" be necessary in the mollification argument?
> >
> > I focus on "absolute continuity" **because that is the assumption of Lemma 3**, which I am claiming is meaningless. There is **NOTHING intrinsically special** about absolute continuity -- it is **certainly insufficient** to get any meaningful quantitative guarantee for stochastic approximation of $\nabla \mu$, this is apparent by considering gradient lipschitz $\mu$'s with arbitrarily bad gradient lipschitz constant.
> >
> > **If** you guys had a different assumption $X$ than absolute continuity, e.g. some lipschitz assumption on $\mu$, leading to some quantitative computation bound, **then I would be talking about assumption $X$ instead**, and I would claim that under $X$, mollification gives quantitative computation bounds that are as good as whatever you claim.

---

### Official Review · Reviewer_n4mm · 2024-11-04

**Soundness:** 3
**Presentation:** 4
**Contribution:** 3
**Rating:** 8
**Confidence:** 3

**Summary:**

This paper presents improved results on gradient estimation by stochastic smoothing. In particular, the density function is assumed to be absolutely continuous as opposed to being differentiable. The authors discuss three ways to reduce variance of the estimated gradients. Finally, the paper presents application of this smoothing on various problem where gradients of non-differentiable functions are required.

This paper builds on existing ideas, but it does present some strong and comprehensive results that are both insightful.

**Strengths:**

The paper is very well written, with ideas clearly conveyed. Mathematical descriptions are precise.

I like the fact that multiple density functions are considered and discussed.

The discussion on variance reduction is insightful.

Many different applications are presented. And at the end, some recommendations are made about the "optimal" densities, as well as on the sampling methods.

**Weaknesses:**

None.

**Questions:**

Can this method be applied to problems where the $ L^1 $ loss is minimized?

---

> ### Author Response · Authors · 2024-11-20
> **Response to Reviewer n4mm**
>
> Dear Reviewer n4mm,
>
> Thank you very much for reviewing our work.
>
> We appreciate you pointing out that our paper "does present some strong and comprehensive results that are both insightful".
>
> Also, thank you for appreciating that our paper "is very well written, with ideas clearly conveyed", and that our "mathematical descriptions are precise".
>
> Further, thanks for appreciating that "multiple density functions are considered and discussed", finding our discussion on variance reduction "insightful", appreciating our "many different applications", and our recommendations "about the 'optimal' densities" and sampling methods.
>
> **Questions:**
>
> > "Can this method be applied to problems where the $L^1$ loss is minimized?"
>
> Yes, it can. For example, in the shortest-path experiment, when smoothing the algorithm, we could have used an $L^1$ loss instead of an $L^2$ loss. For context, in some prior works that built on AlgoVision, $L^1$ was considered in addition to the $L^2$ loss. We can offer to include an additional experiment with the $L^1$ loss in the camera-ready (running many seeds for a scientifically meaningful result is too expensive for the rebuttal phase).

---

### Official Review · Reviewer_NByq · 2024-11-07

**Soundness:** 4
**Presentation:** 4
**Contribution:** 1
**Rating:** 3
**Confidence:** 5

**Summary:**

This work presents methods for stochastic smoothing of non-differentiable functions, motivated by applications where there is the need to optimize non-differentiable functions. Some experiments are provided.

**Strengths:**

.

**Weaknesses:**

The novelty is insufficient. The idea of smoothing using Gaussian perturbations is a well-established technique in 0-th order optimization or non-smooth optimization. The authors present further general smoothing conditions, requiring that the density function be absolutely continuous, but these derivations are mostly standard exercises in mathematical analysis. The variance reduction techniques are also not novel. Overall, my view is that there is very little theoretical novelty.

The authors provide applications and experiments, but I do not think they are strong enough to warrant publication on the basis of the experiments.

**Questions:**

.

---

> ### Author Response · Authors · 2024-11-20
> **Response to Reviewer NByq**
>
> Dear Reviewer NByq,
>
> we are suprised by your very laconic review of our work, considering that you give yourself a confidence rating of 5.
>
> To address your few remarks, we would like to kindly remark that we already clearly point out that "smoothing using Gaussian perturbations is a well-established technique in 0-th order optimization", which is only the starting point of our work.
> Our theory is **not** "mostly standard exercises in mathematical analysis", and indeed Lemma 3 would not even follow from any of the many theory results in the field of mollifiers in mathematical analysis. The strongest result that can be derived from the theory of mollifiers, (and under the additional restriction that the range of $f$ has to be bounded), is to our knowledge the continuity of $f_\epsilon$, but not differentiability. If you insist on your stance, could you kindly provide a reference from the literature for your claim? Moreover, we feel it is necessary to point out that we provide crucial and novel theory even for the case of Gaussian smoothing, e.g., Lemma 8, Lemma 9, and Theorem 7.
>
> Regarding your claim that the variance reduction techniques are not novel, we would also appreciate if you could provide a reference from the literature for QMC or RQMC sampling strategies used in the field of gradient estimation. We clearly point out that the other variance reduction methods have been previously utilized for smoothing (e.g., line 220-222), but include them to enable our experimental comparative study of the variance reduction techniques, e.g., for the very clear result that the popular antithetic sampling performs poor for stochastic smoothing, and our extension by (R)QMC substantially improves the results.
> Our extensive experimental comparative study across 3 dimensions of variance reduction, 6 different distributions, 4 experimental domains, and varying numbers of samples, as well as other extensions like "smoothing of the algorithm" has not been done (even remotely) in any previous work.
> Indeed for all three of the standard benchmark in the differentiable sorting & ranking community (Sec 4.2), the standard benchmark in the differentiable optimizers community (Sec 4.3), as well as an established standard experiment in the differentiable rendering community (Sec 4.4), stochastic smoothing has not been applied in the literature (not even Gaussian smoothing).
> Moreover, cryo-ET (Sec 4.5) has not been previously considered, and was previously in the differentiable physics community considered as a setting that could not even be made differentiable.

---

### Meta-Review · Area_Chair_1D3m · 2024-12-08

**Metareview:**

While the paper aims to address interesting problems, several reviewers have found its theoretical contribution, in light of the existing literature, to be rather minimal, with its impact on machine learning deemed questionable. It has been noted that the theoretical derivations are at times trivial, and the results occasionally lack meaningful applications. Furthermore, the numerical examples fall short of being convincing and fail to complement the shortcomings of the theoretical results.

**Additional Comments On Reviewer Discussion:**

The main criticism directed at this paper stems from its limited theoretical contribution, with reviewers NByq and FXvh both finding that the paper lacks sufficient theoretical merit and impact to warrant acceptance. While the authors have attempted to address some of these criticisms in their rebuttal, it appears that the majority of the concerns remain unresolved.

---

### Decision · Program_Chairs · 2025-01-22

Reject